# Radial Attention: $\mathcal{O}(n \log n)$ Sparse Attention with Energy Decay for Long Video Generation

**Xingyang Li**[*]   **Muyang Li**[*]   **Tianle Cai**   **Haocheng Xi**
**Shuo Yang**   **Yujun Lin**   **Lvmin Zhang**   **Songlin Yang**   **Jinbo Hu**
**Kelly Peng**   **Maneesh Agrawala**   **Ion Stoica**   **Kurt Keutzer**   **Song Han**

MIT   NVIDIA   Princeton   UC Berkeley   Stanford   First Intelligence
https://github.com/mit-han-lab/radial-attention

> "Nothing spreads without loss; every signal, every influence, every attention — decays with distance."          — *Inspired by thermodynamic principles*

Prompt: *A stylish woman walks down a Tokyo street filled with warm glowing neon and animated city signage. She wears a black leather jacket, a long red dress, and black boots, and carries a black purse. She wears sunglasses and red lipstick. She walks confidently and casually. The street is damp and reflective, creating a mirror effect of the colorful lights. Many pedestrians walk about.*

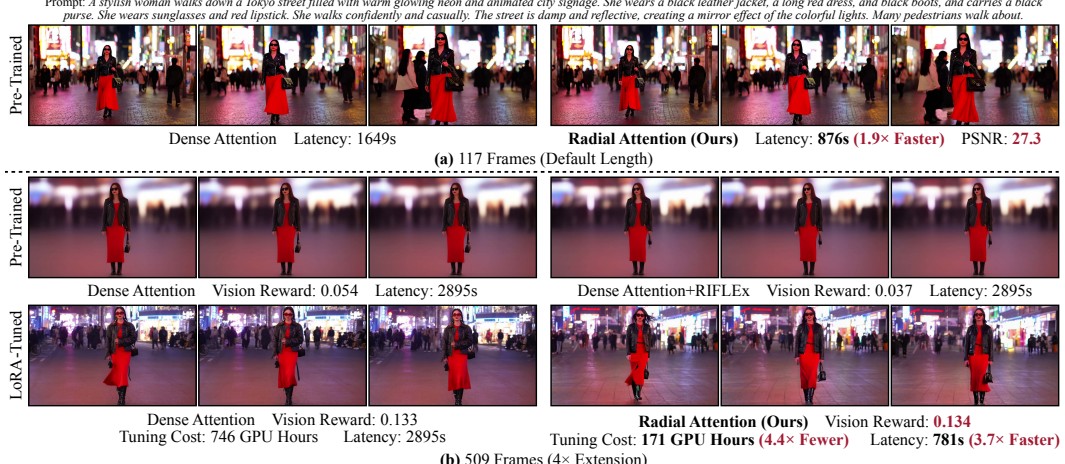

Dense Attention    Latency: 1649s       **Radial Attention (Ours)**    Latency: **876s (1.9× Faster)**    PSNR: **27.3**

**(a)** 117 Frames (Default Length)

Dense Attention    Vision Reward: 0.054    Latency: 2895s       Dense Attention+RIFLEx    Vision Reward: 0.037    Latency: 2895s

Dense Attention    Vision Reward: 0.133       **Radial Attention (Ours)**    Vision Reward: **0.134**
Tuning Cost: 746 GPU Hours    Latency: 2895s       Tuning Cost: **171 GPU Hours (4.4× Fewer)**    Latency: **781s (3.7× Faster)**

**(b)** 509 Frames (4× Extension)

Figure 1: We present Radial Attention, a sparse attention mechanism with $\mathcal{O}(n \log n)$ computational complexity. Radial Attention accelerates pre-trained HunyuanVideo [1] by 1.9× at its default video length while maintaining comparable video quality. When generating 4× longer videos, it reduces tuning costs by up to 4.4× and speeds up inference by up to 3.7× versus dense attention.

## Abstract

Recent advances in diffusion models have enabled high-quality video generation, but the additional temporal dimension significantly increases computational costs, making training and inference on long videos prohibitively expensive. In this paper, we identify a phenomenon we term *Spatiotemporal Energy Decay* in video diffusion models: post-softmax attention scores diminish as spatial and temporal distance between tokens increase, akin to the physical decay of signal or waves over space and time in nature. Motivated by this, we propose *Radial Attention*, a scalable sparse attention mechanism with $\mathcal{O}(n \log n)$ complexity that translates energy decay into exponentially decaying compute density, which is significantly more efficient than standard $\mathcal{O}(n^2)$ dense attention and more expressive than linear attention. Specifically, Radial Attention employs a simple, static attention mask where each token attends to spatially nearby tokens, with the attention window size

---

[*]indicates equal contributions.

39th Conference on Neural Information Processing Systems (NeurIPS 2025).

shrinking with temporal distance. Moreover, it allows pre-trained video diffusion models to extend their generation length with efficient LoRA-based fine-tuning. Extensive experiments show that Radial Attention maintains video quality across Wan2.1-14B, HunyuanVideo, and Mochi 1, achieving up to a 1.9× speedup over the original dense attention. With minimal tuning, it enables video generation up to 4× longer while reducing training costs by up to 4.4× compared to direct fine-tuning and accelerating inference by up to 3.7× compared to dense attention inference. Code is released at https://github.com/mit-han-lab/radial-attention.

# 1 Introduction

Diffusion models have achieved remarkable success in generating high-quality images [2, 3]. Recent advances have extended their capabilities to video generation, producing visually compelling results [4, 5, 6, 7, 1].

However, such advances incur substantial computational costs. Unlike image synthesis, the temporal dimension in video synthesis greatly increases token counts compared to images, and the quadratic scaling of self-attention with context length renders training and inference on long videos computationally prohibitive, restricting model scalability.

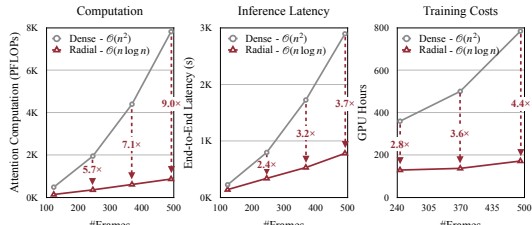

Figure 2: Radial Attention reduces the computational complexity of attention from $\mathcal{O}(n^2)$ to $\mathcal{O}(n \log n)$. When generating a 509-frame 720p video with HunyuanVideo, it reduces the attention computation by 9×, achieves 3.7× speedup, and saves 4.4× tuning costs.

Several prior works tried to mitigate this challenge using sparse attention. For instance, as illustrated in Figure 3(a), Sparse VideoGen (SVG) [8] employs an online profiling strategy that classifies each attention head as either spatial or temporal and then applies the corresponding sparse mask. While this can accelerate inference, it poses challenges during training, especially for longer videos. The profiling may misclassify attention heads on unseen data distributions, whose error can be reinforced during optimization, leading to degraded performance. Other approaches replace the softmax attention with linear alternatives [9, 10], but these often require substantial architectural changes, where modest fine-tuning is typically insufficient to recover the original video quality.

In physics, it is well known that signals and waves lose energy as they propagate through space and time. Inspired by this principle, we observe a similar phenomenon in attention: post-softmax attention scores between tokens diminish as their spatial or temporal distance increases (see Figure 4(b)). We term this phenomenon **Spatiotemporal Energy Decay** and model the decay as an exponential function of both spatial and temporal distances. Based on this model, we unify spatial and temporal attention heads in SVG [8] into **Radial Attention**, a scalable sparse attention mechanism with $\mathcal{O}(n \log n)$ computational complexity (see Figure 2). Radial Attention employs a static sparse attention mask to translate the concept of energy decay into computation density decay. The mask design is simple yet effective: each token attends to others at similar spatial locations, while the attention window shrinks exponentially with temporal distance, as illustrated in Figure 3(b).

Moreover, since Radial Attention only prunes unimportant token relations without modifying the underlying softmax attention mechanism, it enables efficient adaptation of pre-trained video diffusion models to longer sequences using lightweight fine-tuning, such as LoRA [11]. Compared to full-parameter fine-tuning with dense attention, it achieves better video quality, as LoRA focuses on updating parameters most critical for temporal coherence and visual fidelity. The length-extension LoRA is also compatible with existing style LoRAs (see Section 5.2).

When generating videos at the default length, Radial Attention accelerates leading video diffusion models of Wan2.1-14B [7], HunyuanVideo [1] by up to 1.9× speedup. When generating 4× longer videos, Radial Attention reduces tuning costs by up to 4.4× and accelerates inference by up to 3.7× without sacrificing quality. Some visual examples on HunyuanVideo can be found in Figure 1.

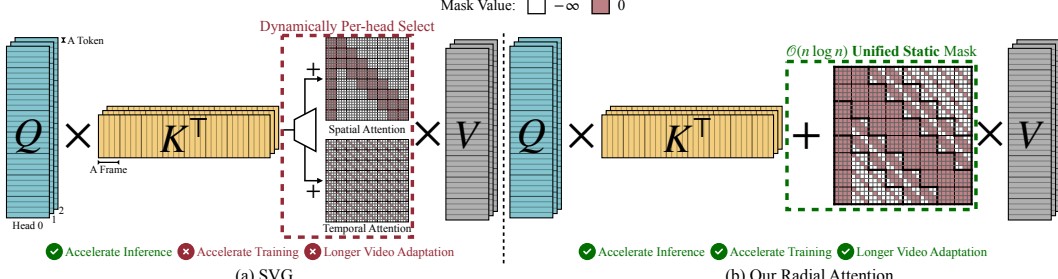

Figure 3: Attention pipelines of SVG [8] and our Radial Attention. Softmax is omitted for clarity. **(a)** SVG dynamically selects either a spatial or temporal attention for each head to speed up inference. However, it does not overcome the original model's length limitation and cannot be trained on unseen distributions like longer videos. **(b)** Our Radial Attention uses a static mask that unifies spatial and temporal attention with $\mathcal{O}(n \log n)$ computational complexity. This static design enables efficient longer-video adaptation.

## 2 Related Work

**Video diffusion models.** Diffusion models have achieved state-of-the-art (SOTA) results in image synthesis [2, 3, 12, 10]. Researchers further extend them to the video domain. Early approaches [13, 14, 15, 16] adapted 2D UNets [2, 17] to handle frame sequences by adding temporal modules. Ever since the advent of Sora [4], the community has largely shifted to use DiT [18] as the backbone. Latte [19] first proposed decoupled spatial and temporal attention for modeling video sequences. To better capture long-range dependencies and jointly model spatial-temporal dynamics, recent SOTA models have adopted 3D dense attention [20, 21, 5, 22, 1, 7, 6]. However, dense attention is computationally intensive due to the additional temporal dimension, and its cost scales quadratically with the number of frames, posing substantial challenges for both training and deployment.

**Efficient video generation.** Many techniques developed to accelerate image diffusion models—such as timestep distillation [23, 24], caching [25, 26], quantization [27, 28, 29], and distributed inference [30, 31, 32]—also apply to video diffusion. However, video models often rely on 3D dense attention, shifting the bottleneck from feedforward to attention layers. Recent works like SageAttention [33, 34, 35, 36] and FlashAttention-3 [37] show that quantizing attention can significantly speed up inference. In large language models (LLMs), sparse attention has been widely applied for acceleration [38, 39, 40, 41, 42, 43, 44, 45, 46]. For instance, Long LoRA [39] combines two local sparse attention patterns with shifting to achieve a global receptive field in video understanding. PowerAttention [45] restricts attention to power-of-two token distances, yielding $\mathcal{O}(n \log n)$ complexity. However, these methods ignore the inherent spatial and temporal structure in video data, making them suboptimal for video generation (see Section 5.2). To better exploit this structure, several video-specific sparse attention methods have been proposed [8, 47, 48, 49]. For example, STA [47] uses sliding 3D windows for local attention, and SVG [8] dynamically selects spatio-temporal patterns for each head. Both improve efficiency but struggle with long videos: STA's fixed receptive field limits long-range dependencies, while SVG's runtime profiling becomes unreliable for unseen long video distributions. In contrast, our Radial Attention employs a static $\mathcal{O}(n \log n)$ pattern across all heads, accelerating both training and inference and enabling efficient extension to longer videos.

**Long video generation.** Due to the quadratic cost of dense attention, training and infernce on long videos remain highly expensive. RIFLEx [50] extends video length by modifying RoPE [51] frequencies to tackle temporal repetition and motion deceleration, allowing $2\times$ extrapolation with the pre-trained models. However, it still suffers from poor video quality (*e.g.*, blurring) when generating longer videos. Dalal *et al.* generate short video segments and stitch them together via test-time training layers [52]. Framepack [53] adopts an autoregressive strategy, generating short clips sequentially based on context frames that are encoded into a fixed number of tokens. Other approaches replace dense attention with linear attention [10, 9, 54, 55, 56, 57, 58, 59], offering faster computation and global receptive fields. However, linear attention struggles to capture local details [60], often degrading quality. Our Radial Attention strikes a middle ground between $\mathcal{O}(n^2)$ dense attention and $\mathcal{O}(n)$ linear attention, achieving $\mathcal{O}(n \log n)$ complexity while preserving the visual fidelity. Moreover, it can be efficiently fine-tuned from existing models using LoRA [11], enabling scalable longer-video generation with minimal overhead.

**Attention with $\mathcal{O}(n \log n)$ complexity.** Preliminary efforts in this direction include Reformer [61], which approximates dense attention via locality-sensitive hashing; H-Transformer [62], which

imposes a hierarchical structure on the attention matrix; Multi-resolution attention [63], which recursively refines high-attention regions; Fast Multipole Attention [64], which adapts the classical fast multipole method; and LogSparse Transformer [65] for time-series forecasting, which restricts each token to attend to $\mathcal{O}(\log n)$ positions per layer. However, these methods are often hardware-unfriendly and exhibit limited scalability. In contrast, our approach employs a simple, block-friendly static mask that scales efficiently while maintaining strong modeling capacity.

## 3    Preliminary

Diffusion models synthesize videos by sampling Gaussian noise $\boldsymbol{X}_T \sim \mathcal{N}(\boldsymbol{0}, \boldsymbol{I})$ in a latent space and progressively denoising it through a neural network to produce a clear latent $\boldsymbol{X}_0$, which is subsequently decoded into the final video using a pre-trained decoder. Compared to images, videos introduce an additional temporal dimension, significantly increasing the number of latent tokens. For instance, generating a 5-second 720p video in HunyuanVideo [1] requires approximately 110K tokens. Excessive latent compression degrades video quality [3], limiting token reduction.

To capture spatiotemporal correlation in video generation, recent models [7, 1, 22, 5] use 3D dense attention, which computes interactions between all token pairs. Given $n$ tokens with embedding dimension $d$, attention is computed as:

$$\text{Attention}(\boldsymbol{Q}, \boldsymbol{K}, \boldsymbol{V}) = \text{softmax}\left(\frac{\boldsymbol{Q}\boldsymbol{K}^\top}{\sqrt{d}}\right)\boldsymbol{V}, \qquad (1)$$

where $\boldsymbol{Q}, \boldsymbol{K}, \boldsymbol{V} \in \mathbb{R}^{n \times d}$ are the query, key, and value matrices. The computation of $\boldsymbol{Q}\boldsymbol{K}^T$ incurs $\mathcal{O}(n^2)$ time and memory complexity. While FlashAttention [66, 67] series reduce memory overhead, the quadratic time complexity remains a bottleneck, especially for long or high-resolution videos. Thus, designing more efficient attention mechanisms is vital for scaling video diffusion models.

To mitigate this computational burden, sparse attention restricts interactions to a subset of token pairs. Formally, this is achieved by adding a sparsity mask $\boldsymbol{M} \in \{-\infty, 0\}^{n \times n}$ to the attention logits:

$$\text{SparseAttention}(\boldsymbol{Q}, \boldsymbol{K}, \boldsymbol{V}) = \text{softmax}\left(\frac{\boldsymbol{Q}\boldsymbol{K}^\top + \boldsymbol{M}}{\sqrt{d}}\right)\boldsymbol{V}. \qquad (2)$$

Entries set to $-\infty$ are ignored in the softmax computation. Various schemes have been proposed to construct the mask. Static methods, such as STA [47], use fixed sparsity patterns but are less expressive. In contrast, dynamic schemes like SVG [8] does dynamic sparse pattern based on input content to improve fidelity. However, dynamic masking introduces online mask decision overhead and does not apply to training. Can we design a static attention pattern that matches the expressiveness of dynamic methods and can also be used in training?

## 4    Method

The key insight of Radial Attention is that attention scores between tokens decay with increasing spatial and temporal distance. This motivates us to allocate computation based on the inherent spatiotemporal correlations. In Section 4.1, we characterize the spatiotemporal energy decay phenomenon in attention. In Section 4.2, we formally define Radial Attention, which translates energy decay into corresponding compute density reduction, enabling speedup. We also analyze its complexity and approximation error, showing that the complexity is $\mathcal{O}(n \log n)$ and the effectiveness of our mask. Finally, in Section 4.3, we show how to extend pre-trained models to longer videos using Radial Attention.

### 4.1    Spatiotemporal Energy Decay in Attention

In Figure 4(a), we show two post-softmax attention maps from HunyuanVideo [1]. Following the terminology in SVG [8], the left map is referred to as spatial attention, where each token primarily attends to nearby tokens within adjacent frames. The right map represents temporal attention, where each token focuses on tokens at the same spatial location across different frames. Figure 4(b) illustrates the attention score distributions of these two maps, along with a third curve of averaged attention scores over multiple heads and diffusion steps. In Figure 4(b1), we show the average attention score between tokens at the same spatial location but with increasing temporal distance. In Figure 4(b2),

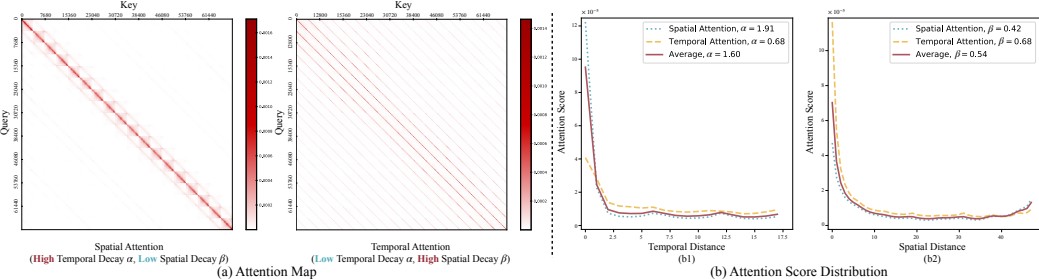

Figure 4: **(a)** Example spatial and temporal attention maps from HunyuanVideo (defined in Section 4.1). **(b)** Attention score distributions. (b1): Average score between tokens at the same spatial location decreases with temporal distance (b2): Average attention score within a frame decreases with spatial distance. *Spatial* and *Temporal Attention* refer to the distributions derived from the corresponding maps in (a). *Average* means averaging over multiple random maps and diffusion steps. The plots indicate that spatial attention shows a high temporal decay and relatively low spatial decay, while temporal attention exhibits the opposite.

we show the average score between tokens in the same frame as spatial distance increases. In both cases, attention scores exhibit a clear decay pattern with increasing distance between the query and key tokens. We refer to this phenomenon as Spatiotemporal Energy Decay. Moreover, regression analysis suggests that this decay closely follows an exponential distribution (see Section 5.3).

Specifically, following the notation from Section 3, assume the video latent consists of $f$ frames, each containing $s$ tokens (in total $n = fs$ tokens). Consider a query token located at the $k_0$-th spatial position of the $i_0$-th frame. The corresponding attention score after applying softmax, denoted by $\boldsymbol{p} \in [0,1]^n$, is given by $\boldsymbol{p} = \text{softmax}(\boldsymbol{Q}_{i_0 s + k_0} \boldsymbol{K}^\top)$. Then there exist constants $\alpha, \beta > 0$ and $C_{\text{rel}} > 0$, for each key token at spatial position $l$ in frame $j$ satisfying

$$p_{js+l} \leq C_{\text{rel}} e^{-\alpha|j-i_0| - \beta|l-k_0|} p_{i_0 s + k_0}. \tag{3}$$

Parameters $\alpha$ and $\beta$ control temporal and spatial decay, respectively. High $\beta$ (strong spatial locality) and low $\alpha$ model temporal attention, while high $\alpha$ and low $\beta$ capture spatial attention, as shown in the empirical plots in Figure 4(b). This motivates our unified sparsity pattern that leverages both spatial and temporal decay in a principled manner.

## 4.2 Radial Attention: Convert the Energy Decay to Compute Density Decay

Radial Attention simulates energy decay through compute density decay to save computation.

**Temporal density decay.** Along the temporal dimension, Radial Attention applies an exponential decay rule: the compute density between tokens in frame $i$ and frame $j$ is $(\frac{1}{2})^{\lfloor \log_2(\max(|i-j|,1)) \rfloor}$. This forms a structured pattern as illustrated in Figure 5(a) with $2\lceil \log_2(\max(f, 2)) \rceil - 1$ diagonal bands centered on the main diagonal (band 0). Bands above and below the diagonal are indexed as $1, 2, 3, \ldots$ and $-1, -2, -3, \ldots$, respectively. Each band's width doubles relative to the previous one, ensuring that the total computation per band remains bounded by a constant. The attention from tokens in frame $i$ to frame $j$ lies in band $\text{sign}(j - i) \cdot \lfloor \log_2 \max(|i-j|, 1) \rfloor$. The central band (band 0) retains 100% compute density, while each successive band moving outward has half the compute density of the preceding one – producing a radial decay effect with progressively lighter colors.

**Spatial density decay.** As observed in Figure 4 and formalized in Equation 3, most attention energy is concentrated on tokens at similar spatial locations across frames. We preserve these high-energy interactions, which yield diagonal-like structures within each frame-to-frame attention block. Due to temporal decay, the computed diagonal width of these blocks shrinks as the temporal distance between frames increases. Specifically, as shown in Figure 5(b), the diagonal width for attention between frame $i$ and frame $j$ is given by $\lfloor \frac{s}{2^{\lfloor \log_2 \max(|i-j|,1) \rfloor}} \rfloor$. If it falls below 1, instead of further narrowing the diagonal, we reduce the frequency of diagonals. Specifically, we only retain diagonals in those blocks where $|i-j| \bmod \lceil \frac{2^{\lfloor \log_2 \max(|i-j|,1) \rfloor}}{s} \rceil = 0$ to keep the same amortized attention density decay.

**Formal definition.** Here we formally define the Radial Attention mask. We construct a 4D attention mask $\tilde{\boldsymbol{M}} \in \{-\infty, 0\}^{f \times f \times s \times s}$, where each element $\tilde{M}_{i,j,k,l} = 0$ indicates that the token at spatial position $k$ in frame $i$ is permitted to attend to the token at position $l$ in frame $j$. Conversely, $\tilde{M}_{i,j,k,l} =$

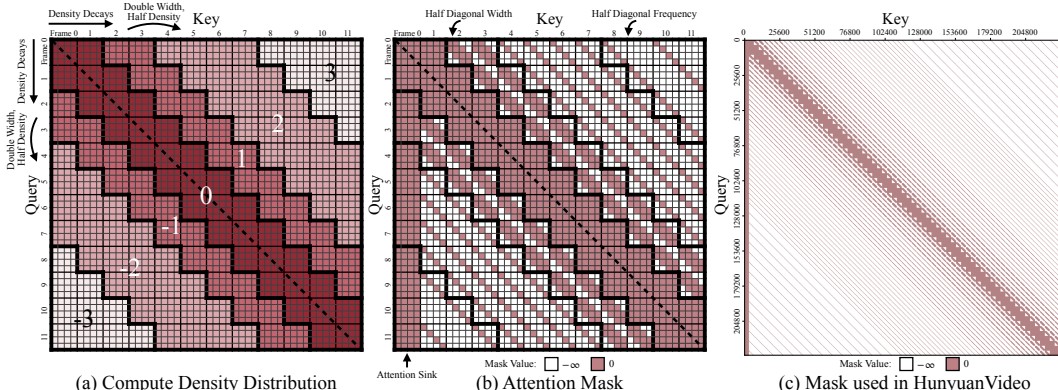

| (a) Compute Density Distribution | (b) Attention Mask | (c) Mask used in HunyuanVideo |

Figure 5: **(a)** The compute density pattern. The attention map is divided into $2\lceil\log_2(\max(f,2))\rceil - 1$ bands (here, the number of frames $f = 12$) based on the temporal distance between tokens. The central band has full compute density, while each successive outer band has half the density of the previous one. Except for band $\pm 1$, each band also doubles the diagonal width of its predecessor. **(b)** The corresponding attention mask for (a). The compute density is reflected in the compute diagonal width of each frame-to-frame block. When the diagonal width drops below 1, we reduce the frequency of diagonals. We additionally add an attention sink. **(c)** An example mask used in HunyuanVideo, illustrating the final sparsity pattern in practice.

$-\infty$ denotes that attention between the token pair is suppressed. The mask is constructed according to:

$$
\tilde{M}_{i,j,k,l} = \begin{cases} 0, & \text{if } 2^{\lfloor\log_2 \max(|i-j|,1)\rfloor} \leq s \text{ and } |k-l|+1 \leq \frac{s}{2^{\lfloor\log_2 \max(|i-j|,1)\rfloor}} \\ 0, & \text{if } |i-j| \bmod \lceil \frac{2^{\lfloor\log_2 \max(|i-j|,1)\rfloor}}{s}\rceil = 0 \text{ and } k = l \\ -\infty. & \text{otherwise} \end{cases} \tag{4}
$$

The final attention mask $M \in \{-\infty, 0\}^{n \times n}$ used in the attention operation of Equation 2 is obtained by flattening frame and spatial indices: $M_{is+k,js+l} = \tilde{M}_{i,j,k,l}$. For better quality, we incorporate an attention sink [38, 8] as the first frame's attention is crucial. Figure 5(c) shows a example mask we use in HunyuanVideo for generating a 253-frame 720p video. This strategy keeps spatial interactions with high temporal proximity while using sparse sampling for distant frames to maintain efficiency.

**Relation to SVG.** Radial Attention unifies spatial and temporal attention in SVG [8] using a single attention mask. Specifically, the central band (band 0 in Figure 5(a)) of our mask already captures dense spatial interactions, effectively subsuming the spatial attention in SVG. For temporal attention, SVG overlooks temporal decay, allocating unnecessary computation to distant frames with low relevance. In contrast, Radial Attention reduces attention to these regions and reallocates the budget toward tokens nearer in time, achieving both improved efficiency and enhanced modeling.

**Complexity analysis.** The computational cost of our method is proportional to the number of zeros in the attention mask $\tilde{M}$. When the number of frames $f$ is large, we derive the following upper bound:

$$
\#\text{zeros in } \tilde{M} \leq \underbrace{4s^2 f}_{\text{central band and sink}} + \underbrace{\sum_{r=1}^{\lfloor\log_2 s\rfloor} 2^{r+1} f \frac{2s^2}{2^r}}_{\text{diagonal width}\geq 1} + \underbrace{\sum_{r=\lfloor\log_2 s\rfloor+1}^{\lceil\log_2 f\rceil-1} 2^{\lfloor\log_2 s\rfloor+1} fs}_{\text{diagonal width}<1} \tag{5}
$$

$$
\leq 4s^2 f \log_2 f = 4sn(\log_2 n - \log_2 s). \tag{6}
$$

A detailed derivation of Equation 5 can be found in Appendix A.1. From Equation 6, we find that for long videos (i.e., large $f$) with fixed resolution $s$, the computational complexity scales as $\mathcal{O}(n \log n)$. Empirical results on HunyuanVideo, shown in Figure 2, confirm this trend. Notably, our Radial Attention reduces attention computation by 9× compared to dense attention for 4× longer videos.

**Error analysis.** Following Equation 3, we derive an error bound for the attention score corresponding to a query token at position $k_0$ in the $i_0$-th frame. $\tilde{p} = \text{softmax}(Q_{i_0 s+k_0} K^\top + \tilde{M}_{i_0 s+k_0})$ denotes the masked attention score. The $\ell_1$ attention error of our approximated attention is bounded as follows:

$$
\|\tilde{p} - p\|_1 \leq C_{\text{rel}} \left[ \frac{8 e^{-\beta\left(\frac{s}{2}+1\right)}}{(1-e^{-\alpha})(1-e^{-\beta})} + 4 \frac{1+e^{-\beta}}{1-e^{-\beta}} \frac{e^{-\alpha(s+1)}}{1-e^{-\alpha}} \right] = O(C_{\text{rel}} e^{-\min(\beta/2,\alpha)s}). \tag{7}
$$

Table 1: Quantitative results at the default video length. Under the same computation budget, our method consistently outperforms STA and PA in PSNR, SSIM, and LPIPS, matches the video fidelity of SVG, and achieves 1.8× speedup on HunyuanVideo and Wan2.1-14B on a single H100 GPU.

| Model | Method | PSNR (↑) | SSIM (↑) | LPIPS (↓) | Vision Reward (↑) | PFLOPs | Latency (s) | Speedup |
|---|---|---|---|---|---|---|---|---|
| Hunyuan Video (117 frames) | Original | – | – | – | 0.141 | 612 | 1649 | – |
| | STA (FA3) | 26.7 | 0.866 | 0.167 | 0.132 | 331 | 719 | 2.29× |
| | PA | 22.1 | 0.764 | 0.256 | 0.140 | 339 | 1002 | 1.65× |
| | SVG | 27.2 | **0.895** | **0.114** | **0.144** | 340 | 867 | 1.90× |
| | Ours | **27.3** | 0.886 | **0.114** | 0.139 | 339 | 876 | 1.88× |
| Wan2.1-14B (69 frames) | Original | – | – | – | **0.136** | 560 | 1630 | – |
| | STA (FA3) | 22.9 | 0.830 | 0.171 | 0.132 | 322 | 812 | 2.01× |
| | PA | 22.4 | 0.790 | 0.176 | 0.126 | 324 | 978 | 1.67× |
| | SVG | 23.2 | 0.825 | 0.202 | 0.114 | 324 | 949 | 1.71× |
| | Ours | **23.9** | **0.842** | **0.163** | 0.128 | 323 | 917 | 1.77× |

Proof details are provided in Appendix A.2. As Equation 7 shows, the error decreases exponentially with larger decay rates $\alpha$ and $\beta$. In Section 5.3, we further empirically compare this error bound to that of SVG, showing that Radial Attention achieves smaller errors, thereby validating its effectiveness.

**Hardware-friendly block sparsity.** To ensure efficient execution on modern hardware, attention is computed over $128 \times 128$ blocks rather than individual $1 \times 1$ tokens [68, 8, 40, 43, 44, 66].

### 4.3 Low-Rank Adaptation for Long Videos

Although we employ an efficient attention mechanism, the pre-trained model was originally trained on short videos. Recent works [50] have explored training-free methods for extending generation to longer videos, but their performance remains limited due to length distribution mismatch. Training directly on long videos, meanwhile, is computationally prohibitive. Radial Attention alleviates this challenge by reducing the training time complexity to $\mathcal{O}(n \log n)$. Importantly, it preserves critical inter-token relations in the softmax attention, allowing the original pre-trained weights to remain largely intact. Thus, only minimal fine-tuning is required. To further minimize training overhead, we incorporate low-rank adapters (LoRA) [11, 39] into the attention mechanism. Specifically, LoRA is applied to the query, key, value, and output projections of the attention layers, enabling efficient fine-tuning with significantly reduced memory and computational costs. Empirically, we find that LoRA fine-tuning with Radial Attention not only minimizes overhead but also improves video quality by refining only the most critical weights and attention more effectively. See Section 5.3 for detailed results.

## 5 Experiments

### 5.1 Setups

**Models.** We benchmark our method on three popular text-to-video diffusion models: Mochi 1 [22], HunyuanVideo [1], and Wan2.1 [7], which contain 10, 13, and 14 billion parameters, respectively. Mochi 1 can generate up to a 5-second video with 480p resolution and 162 frames. HunyuanVideo can generate up to a 5-second video with 720p resolution and 125 frames. Wan2.1-14B can generate up to a 5-second video with 720p and 81 frames.

**Benchmarks.** We use Vision Reward [69] (higher is better) to approximate the human rating of the generated videos. For pre-trained models evaluated at their default video lengths, we further report PSNR and SSIM to quantify numerical similarity, and LPIPS [70] to assess perceptual differences between the outputs of the original models and the benchmarked methods. For longer-video generation, we additionally use VBench-long[71] to evaluate our fine-tuned models. Specifically, we report metrics of subject consistency, aesthetic quality, and imaging quality, where the original models exhibit notable degradation.

**Baselines.** We compare Radial Attention against the following methods:

- **SVG [8]:** Accelerates video models with sparse attention by dynamically classifying attention heads as spatial or temporal and applying corresponding masks.

- **Spatial/Temporal:** The respective attention masks used in SVG's spatial and temporal heads, as described in Section 4.1.

- **STA [47]:** Applies sliding window attention to capture spatially and temporally local dependencies.

Table 2: Quantitative results at the extended (2× and 4×) video lengths. With minimal fine-tuning, our method maintains the quality regarding Vision Reward and multiple VBench dimensions (Subject Consistency, Aesthetic Quality, and Image Quality) when the length grows. It also achieves high sparsity, reducing training costs by up to 4.4× and delivering up to 3.7× inference speedup.

| Model | #Frames | Method | Sparsity | Training Time (h) | Training Speedup | Inference Time (s) | Inference Speedup | Vision Reward (↑) | VBench S.C. | A.Q. | I.Q. |
|---|---|---|---|---|---|---|---|---|---|---|---|
| Hunyuan Video | 125 (1×) | Original | 0.00% | – | – | 225 | – | 0.119 | 0.959 | 0.643 | 0.672 |
| | 253 (2×) | Original | 0.00% | – | – | 797 | 1.00× | 0.122 | 0.953 | 0.603 | 0.611 |
| | | RIFLEx | 0.00% | – | – | 797 | 1.00× | **0.128** | 0.969 | 0.622 | 0.614 |
| | | Spatial | 80.5% | 16.0 | 2.81× | 335 | 2.38× | 0.054 | **0.979** | 0.607 | 0.670 |
| | | Temporal | 80.7% | 16.2 | 2.78× | 338 | 2.36× | 0.104 | 0.963 | 0.620 | 0.658 |
| | | Long LoRA | 80.6% | 16.6 | 2.71× | 363 | 2.20× | 0.112 | 0.958 | 0.620 | **0.685** |
| | | PA [45] | 80.4% | 16.7 | 2.69× | 334 | 2.39× | 0.109 | 0.967 | 0.608 | 0.653 |
| | | SANA | – | 12.8 | 3.52× | 285 | 2.80× | -0.205 | 0.907 | 0.300 | 0.442 |
| | | Full | 0.00% | 45.0 | 1.00× | 797 | 1.00× | 0.124 | 0.955 | 0.616 | 0.648 |
| | | Ours | 80.8% | 16.2 | **2.78×** | 339 | **2.35×** | 0.126 | 0.968 | **0.623** | 0.663 |
| | 509 (4×) | Original | 0.00% | – | – | 2895 | 1.00× | 0.054 | 0.988 | 0.545 | 0.451 |
| | | RIFLEx | 0.00% | – | – | 2895 | 1.00× | 0.037 | **0.989** | 0.539 | 0.456 |
| | | Spatial | 88.3% | 20.7 | 4.52× | 755 | 3.83× | 0.112 | 0.922 | 0.598 | 0.664 |
| | | Temporal | 88.2% | 21.1 | 4.44× | 774 | 3.74× | 0.083 | 0.972 | 0.597 | 0.646 |
| | | Long LoRA | 88.4% | 20.9 | 4.48× | 803 | 3.61× | 0.130 | 0.936 | 0.618 | **0.689** |
| | | PA [45] | 88.2% | 21.8 | 4.29× | 766 | 3.78× | 0.128 | 0.950 | 0.590 | 0.648 |
| | | Full | 0.00% | 93.6 | 1.00× | 2895 | 1.00× | 0.133 | 0.977 | 0.590 | 0.635 |
| | | Ours | 88.3% | 21.4 | **4.37×** | 781 | **3.71×** | **0.134** | 0.973 | **0.623** | 0.672 |
| Mochi 1 | 163 (1×) | Original | 0.00% | – | – | 112 | – | 0.071 | 0.973 | 0.623 | 0.672 |
| | 331 (2×) | Original | 0.00% | – | – | 302 | 1.00× | 0.040 | 0.937 | 0.551 | 0.466 |
| | | Spatial | 76.1% | 8.57 | 1.75× | 186 | 1.62× | 0.088 | 0.935 | 0.596 | 0.595 |
| | | Temporal | 76.3% | 8.54 | 1.76× | 189 | 1.60× | 0.075 | 0.936 | 0.591 | 0.593 |
| | | Long LoRA | 76.0% | 9.07 | 1.65× | 210 | 1.44× | 0.095 | 0.950 | 0.596 | **0.630** |
| | | PA [45] | 77.8% | 8.53 | 1.76× | 183 | 1.65× | 0.101 | 0.946 | 0.610 | 0.626 |
| | | SANA | – | 8.22 | 1.82× | 166 | 1.82× | -0.201 | 0.905 | 0.334 | 0.568 |
| | | Full | 0.00% | 15.0 | 1.00× | 302 | 1.00× | 0.095 | 0.923 | 0.610 | 0.594 |
| | | Ours | 76.4% | 8.43 | **1.78×** | 185 | **1.63×** | **0.110** | **0.951** | 0.615 | 0.602 |
| | 667 (4×) | Original | 0.00% | – | – | 992 | 1.00× | -0.091 | 0.916 | 0.383 | 0.322 |
| | | Spatial | 85.2% | 17.4 | 2.83× | 382 | 2.60× | 0.091 | 0.930 | 0.611 | 0.585 |
| | | Temporal | 85.4% | 17.6 | 2.80× | 393 | 2.52× | 0.028 | 0.931 | 0.556 | 0.536 |
| | | Long LoRA | 86.0% | 19.0 | 2.59× | 426 | 2.33× | 0.086 | 0.944 | 0.584 | 0.543 |
| | | PA [45] | 86.5% | 17.3 | 2.84× | 381 | 2.60× | 0.107 | 0.956 | **0.633** | **0.650** |
| | | Full | 0.00% | 49.2 | 1.00× | 992 | 1.00× | 0.099 | 0.934 | 0.613 | 0.613 |
| | | Ours | 85.5% | 17.4 | **2.83×** | 386 | **2.57×** | **0.113** | **0.958** | 0.618 | 0.638 |
| Wan2.1 -14B | 81 (1×) | Original | 0.00% | – | – | 1630 | – | 0.135 | 0.973 | 0.623 | 0.672 |
| | 161 (2×) | Original | 0.00% | – | – | 5735 | 1.00× | 0.109 | 0.946 | 0.598 | 0.614 |
| | | Full | 0.00% | 28.0 | 1.00× | 5735 | 1.00× | **0.150** | 0.966 | 0.590 | **0.689** |
| | | Ours | 73.6% | 14.5 | **1.93×** | 2847 | **2.01×** | 0.145 | **0.981** | **0.607** | 0.677 |

- **PowerAttention (PA) [45]:** A sparse attention mechanism with $\mathcal{O}(n \log n)$ complexity for LLMs, attending only to tokens at power-of-two distances.

- **LongLoRA [39]:** Uses shifted local attention to efficiently extend the context window of LLMs.

- **SANA[10]:** An efficient diffusion model backbone with linear attention. We replace softmax attention with SANA's for adapting to longer videos.

- **RIFLEx[50]:** Training-free video length extrapolation by adjusting the frequency of RoPE [51].

**Implementation details.** In terms of Radial Attention implementation, we use FlashInfer [72] for inference and Block-Sparse-Attention [73] with FlashAttention-2 [67] backend during training. For default-length inference, we evaluate HunyuanVideo on 117 frames at 768p resolution (768×1280), and Wan2.1 on 69 frames at the same resolution. Following SVG, we apply dense attention during the first 12 steps as a warm-up phase for all models. Additionally, we keep dense attention in the first DiT block to maintain quality. We measure all the latencies with a single NVIDIA H100 GPU.

For longer-video generation, we fine-tune the model with videos that are 2∼4× longer than the default length from OpenVid-1M [74]. Specifically, we sample 2k top-scoring videos in aesthetic and motion scores for each extended length. We use 8 H100 GPUs for training, which takes around 16∼21 hours for HunyuanVideo, 8∼17 hours for Mochi 1, and 15 hours for Wan 2.1. Inference latency for Wan 2.1 is measured on a single H100, while HunyuanVideo and Mochi 1 are evaluated using 8 H100s. See Appendix B for more details.

*Prompt: A close-up shot captures a cluster of plump, dewy grapes, glistening under soft studio lighting as they slowly rotate on a sleek, reflective table. The grapes, varying in shades of deep purple and rich green, showcase their smooth, taut skins and tiny droplets of moisture.*

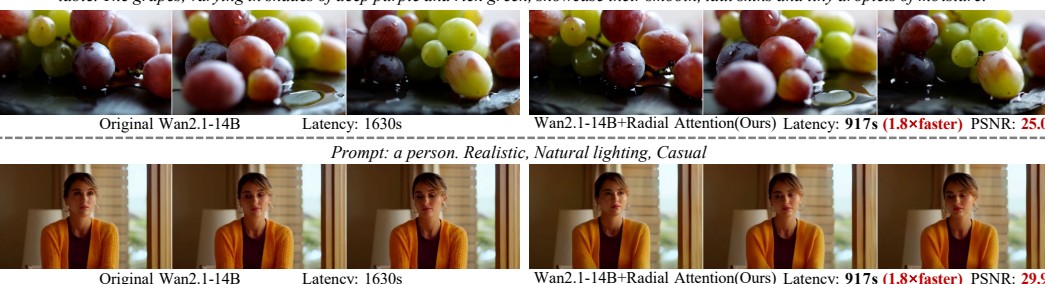

Original Wan2.1-14B   Latency: 1630s   Wan2.1-14B+Radial Attention(Ours) Latency: **917s (1.8×faster)** PSNR: **25.0**

*Prompt: a person. Realistic, Natural lighting, Casual*

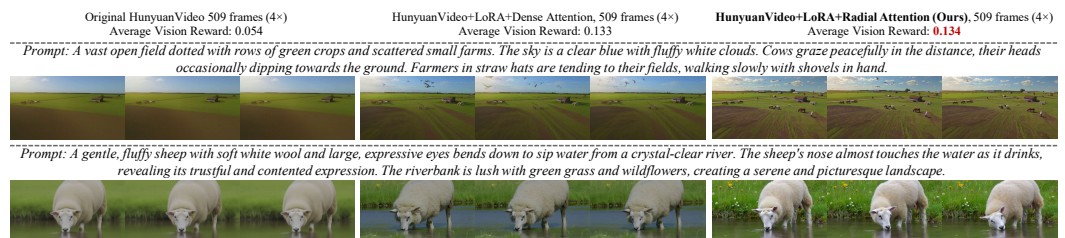

Original Wan2.1-14B   Latency: 1630s   Wan2.1-14B+Radial Attention(Ours) Latency: **917s (1.8×faster)** PSNR: **29.9**

Figure 6: Examples of videos generated by Radial Attention and the original Wan2.1-14B in the default video length. Radial Attention mirrors the video quality of the original model.

Original HunyuanVideo 509 frames (4×)   HunyuanVideo+LoRA+Dense Attention, 509 frames (4×)   **HunyuanVideo+LoRA+Radial Attention (Ours)**, 509 frames (4×)
Average Vision Reward: 0.054    Average Vision Reward: 0.133     Average Vision Reward: **0.134**

*Prompt: A vast open field dotted with rows of green crops and scattered small farms. The sky is a clear blue with fluffy white clouds. Cows graze peacefully in the distance, their heads occasionally dipping towards the ground. Farmers in straw hats are tending to their fields, walking slowly with shovels in hand.*

*Prompt: A gentle, fluffy sheep with soft white wool and large, expressive eyes bends down to sip water from a crystal-clear river. The sheep's nose almost touches the water as it drinks, revealing its trustful and contented expression. The riverbank is lush with green grass and wildflowers, creating a serene and picturesque landscape.*

Figure 7: Visual comparison of HunyuanVideo with 4× length extension (509 frames). LoRA fine-tuned models using Radial Attention achieve higher vision rewards, outperforming dense attention baselines, while achieving a 3.7× speedup and reducing tuning costs by 4.4×.

## 5.2 Main Results

**Training-free inference acceleration.** Table 1 presents a quantitative comparison of Radial Attention against three strong sparse attention baselines on HunyuanVideo [1] and Wan2.1-14B [7] at their default generation lengths. Corresponding visual results are provided in Figure 6 and Appendix C.1. Under the same compute budget (measured in PFLOPs), Radial Attention preserves the video quality of dense attention while consistently outperforming STA and PA on similarity metrics (PSNR, SSIM, LPIPS), and matches the quality of SVG. While PA shares a similar $\mathcal{O}(n \log n)$ complexity with our design, it ignores the spatio-temporal locality inherent in video data, making it suboptimal in practice.

Regarding efficiency, we adopt the same system optimization used in SVG [8]. Specifically, on a single H100, our Radial Attention achieves 1.9× and 1.8× end-to-end speedups for HunyuanVideo and Wan 2.1, respectively, matching the theoretical compute budget savings (1.8× and 1.7× fewer PFLOPs). Although STA yields slightly higher speedup by using FlashAttention-3 (FA-3) [37], it suffers from noticeably degraded visual quality. Our current implementation uses FA2 [67]. Upgrading to FA3 is orthogonal to our algorithm and is left as future work.

**Long video generation.** Table 2 provides results for video generation at 2× and 4× the original lengths, with visualizations available in Figure 7 and Appendix C.2. For Wan2.1-14B, only 2× extrapolation is reported due to its significantly higher computational and memory costs. To ensure fairness, all sparse attention baselines use similar sparsity ratios.

When generating longer videos, the original models without further tuning exhibit significant quality degradation, especially for 4× video length extension. While RIFLEx improves performance at 2× length extrapolation, its quality deteriorates beyond that, indicating limited extension capability. Spatial and temporal sparse attentions suffer from limited reception fields; on the other hand, LongLoRA and PA, though with a global reception field, fail to capture spatial-temporal correlations, resulting in degraded quality. Interestingly, PA exhibits a large gain in Vision Reward after fine-tuning, suggesting that its original sparse pattern misaligns with the pre-trained attention distribution. Fine-tuning allows the model to adapt to the imposed attention sparsity, improving alignment and quality. SANA, which replaces softmax attention with linear attention, requires massive retraining and fails under fine-tuning-based video length extension. In contrast, Radial Attention achieves quality on par with LoRA fine-tuned dense attention models. Notably, it even slightly improves the Vision Reward over the pre-trained model at the default video length.

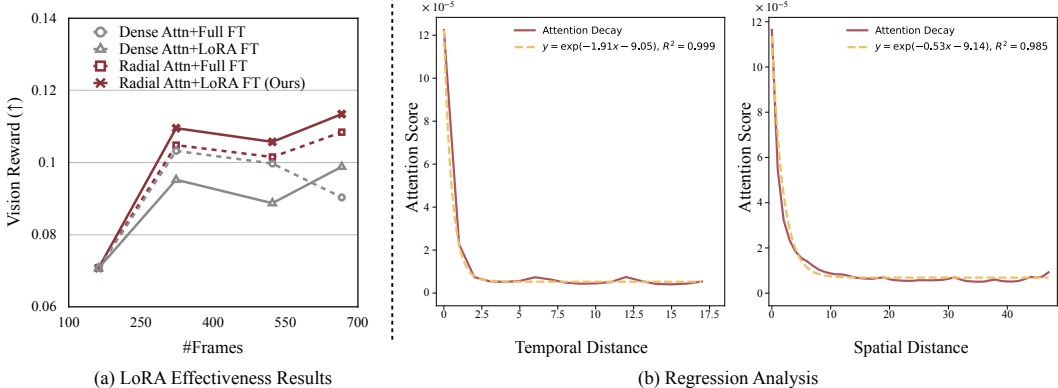

(a) LoRA Effectiveness Results          (b) Regression Analysis

Figure 8: **(a)** Radial Attention outperforms dense attention in generation quality. When combined with LoRA, it further improve the quality while significantly reducing training costs. **(b)** We model the attention decay curves using the exponential function $y = \exp(-ax + b)$. It fits the data well, achieving $R^2 > 0.985$.

Thanks to $\mathcal{O}(n \log n)$ complexity, Radial Attention delivers substantial inference and training speedups over dense attention, as detailed in Table 2 and Figure 2. For instance, when generating 4× longer videos, we can save up to 4.4× training costs and get up to 3.7× inference speedup.

**Compatibility with existing LoRAs.** A key advantage of Radial Attention is its seamless compatibility with pre-trained task-specific LoRAs (*e.g.*, artistic style transfer). We observe that Radial Attention is compatible with existing style LoRAs in both the default length settings, and the longer video settings by directly merging the LoRA weights of Radial Attention in long videos and the style LoRAs. Visualizations and further analysis can be found in Appendix C.3.

### 5.3 Ablation Study & Analyses

**Effectiveness of Low-Rank Adaptation.** Figure 8(a) compares Vision Reward between full fine-tuning and LoRA as video length increases. For dense attention, LoRA fine-tuning lags behind full fine-tuning until 4× length extension. However, with our proposed Radial Attention, LoRA fine-tuning matches or even outperforms full fine-tuning, suggesting that Radial Attention not only scales better computationally, but also makes the model easier to adapt to longer-video generation.

**Attention error.** We evaluate the average attention output Mean Squared Error (MSE, lower is better) of Radial Attention on Wan2.1-14B, comparing it to SVG [8] and STA [47]. Radial Attention achieves an MSE of $3.9 \times 10^{-3}$, significantly lower than $4.4 \times 10^{-3}$ for SVG and $1.5 \times 10^{-2}$ for STA, demonstrating the effectiveness of our mask in preserving attention fidelity.

**Regression results.** We perform regression analysis using the model $y = \exp(-ax + b)$ to fit the average attention decay curves in Figure 4. As illustrated in Figure 8, the fitted curves achieve an $R^2$ value of over 0.985, indicating that the exponential functions can well model the decay.

**More ablations on Radial Attention design choice.** Please refer to Appendix D for more details.

## 6 Conclusion & Discussion

In this work, we propose Radial Attention, an $\mathcal{O}(n \log n)$ sparse attention for efficient long video generation. We observe *Spatiotemporal Energy Decay* in video diffusion models, which motivates a unified attention pattern with sub-quadratic complexity. At default video length, Radial Attention achieves up to a 1.9× speedup with high fidelity. For videos up to 4× longer, Radial Attention preserves quality and delivers up to 4.4× and 3.7× speedups in training and inference, respectively, with minimal LoRA fine-tuning. This work contributes toward scalable, high-quality video generation and offers a foundation for efficient long-range attention in broader sequence modeling tasks.

**Limitations.** The assumption of exponential decay for attention scores (Equation 3) simplifies the complex spatiotemporal dependencies in natural video data. While aiding theoretical analysis, future work could improve efficiency and performance by more deeply understanding and modeling the underlying data structure. As shown in Equation 6, our method still has quadratic complexity with respect to resolution. Future work should explore more efficient attention mechanisms and pre-training strategies, as in NSA [75] and MoBA [76], to better support long, high-resolution videos.

## Acknowledgments

We thank MIT-IBM Watson AI Lab, National Science Foundation, Hyundai, and Amazon for supporting this research.

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

# A   Derivations and Proofs

## A.1   Complexity

In this section, we provide the detailed complexity derivation in Equation 5, which scales as $\mathcal{O}(n \log n)$. Since the computational cost of masked attention is proportional to the number of zeros in the attention mask $\tilde{M}$, we only need to derive an $\mathcal{O}(n \log n)$ upper bound for the latter.

**Central band&attention sink.** Firstly, recall from Figure 5(a) that we apply dense attention on these frame-to-frame attention blocks within the central band and attention sink. The attention sink refers to the pattern that every token attends to all tokens in the first frame. Using the same notation as in Section 4, where $n$ is the total number of tokens, $s$ is the number of tokens per frame, and $f$ is the number of frames (so $n = fs$), we define the attention mask for this region as $\tilde{M}^{(1)} \in \{-\infty, 0\}^{f \times f \times s \times s}$:

$$\tilde{M}^{(1)}_{i,j,k,l} = \begin{cases} 0, & \text{if } |i - j| \leq 1 \text{ or } j = 0 \\ -\infty. & \text{otherwise} \end{cases} \tag{8}$$

Here, $\tilde{M}^{(1)}_{i,j,k,l}$ indicates whether the $k$-th token in frame $i$ is allowed to attend to the $l$-th token in frame $j$, with 0 denoting allowed attention and $-\infty$ indicating disallowed attention. Since the attention sink spans $f$ blocks and the central band includes at most $3f$ blocks, the total number of nonzero entries in this region is bounded by:

$$\#\text{zeros in } \tilde{M}^{(1)} \leq 4 \cdot f \cdot s^2 = 4s^2 f. \tag{9}$$

**Bands with diagonal width $\geq 1$.** The second part is those bands with diagonal width $\geq 1$, except the central band. The mask for this region can be defined as $M^{(2)} \in \{-\infty, 0\}^{f \times f \times s \times s}$:

$$\tilde{M}^{(2)}_{i,j,k,l} = \begin{cases} 0, & \text{if } 2^{\lfloor \log_2 \max(|i-j|,1) \rfloor} \leq s \text{ and } |k - l| + 1 \leq \frac{s}{2^{\lfloor \log_2 \max(|i-j|,1) \rfloor}} \\ -\infty. & \text{otherwise} \end{cases} \tag{10}$$

Thus, since there are at most $\lfloor \log_2 s \rfloor$ bands in this region, the number of zeros in these bands is bounded by:

$$\#\text{zeros in } \tilde{M}^{(2)} \leq \sum_{r=1}^{\lfloor \log_2 s \rfloor} \underbrace{2^{r+1} sn}_{\text{area bound for band } \pm r} \cdot \underbrace{2/2^r}_{\text{compute density bound of band } \pm r} \tag{11}$$

$$\leq \sum_{r=1}^{\lfloor \log_2 s \rfloor} \frac{2^{r+2} s^2 f}{2^r} \tag{12}$$

$$= 4s^2 f \cdot \lfloor \log_2 s \rfloor. \tag{13}$$

**Bands with diagonal width $< 1$.** The last part is those bands with $\frac{s}{2^{\lfloor \log_2 \max(|i-j|,1) \rfloor}} < 1$, where we reduce the frequency of diagonals. The mask for this region $M^{(3)} \in \{-\infty, 0\}^{f \times f \times s \times s}$ is given by:

$$\tilde{M}^{(3)}_{i,j,k,l} = \begin{cases} 0, & \text{if } |i - j| \bmod \lceil \frac{2^{\lfloor \log_2 \max(|i-j|,1) \rfloor}}{s} \rceil = 0 \text{ and } k = l \\ -\infty. & \text{otherwise} \end{cases} \tag{14}$$

Since there are at most $(\lceil \log_2 f \rceil - 1) - (\lfloor \log_2 s \rfloor + 1)$ bands satisfying this condition, we have the number of zeros in these bands bounded by:

$$\#\text{zeros in } \tilde{M}^{(3)} \leq \sum_{r=\lfloor \log_2 s \rfloor + 1}^{\lceil \log_2 f \rceil - 1} \underbrace{2^{\lfloor \log_2 s \rfloor + 1}}_{\text{number of diagonals}} \cdot \underbrace{n}_{\text{area bound of each diagonal}} \tag{15}$$

$$\leq (\lfloor \log_2 f \rfloor - \lfloor \log_2 s \rfloor) 4s^2 f. \tag{16}$$

Combining Equation 9, Equation 13, and Equation 16 together, we have the aggregate upper bound of the number of zeros in Radial Attention's mask:

$$\text{\# of zeros in } \tilde{M} \leq 4s^2 f \cdot \lfloor \log_2 f \rfloor \leq 4s \cdot n(\log_2 n - \log_2 s), \tag{17}$$

which scales $\mathcal{O}(n \log n)$ with the number of frames $f$ for long video generation.

## A.2 Error Bound

The design of Radial Attention is inspired by the spatial-temporal structure in video. In this section, we formulate this intuition by theoretically bounding the asymptotic approximation error of Radial Attention with respect to the decay speed of the attention value in the spatial and temporal dimensions.

We focus on bounding the approximation error of a single row of the attention matrix. We fix a reference query token at position $k_0$ of frame $i_0$, and write the unnormalized row of the attention matrix as

$$a_{j,l} \;=\; \exp(\boldsymbol{Q}_{i_0 s + k_0} \boldsymbol{K}_{js+l}^\top).$$

where $\boldsymbol{Q}_{i_0 s + k_0}$ refers to the query vector at position $k_0$ in frame $i_0$, $\boldsymbol{K}_{js+l}$ refers to the key vector at position $l$ in frame $j$, and $s$ refers to the number of tokens per frame.

### Assumptions

(A1) **Relative exponential decay.** To capture the intuition that the closer frames have a stronger correlation and each token typically attends to tokens in other frames at similar spatial positions, we assume there exist $C_{\text{rel}} > 0$ and $(\alpha, \beta) > 0$ such that

$$0 \;\leq\; a_{j,l} \;\leq\; C_{\text{rel}}\, e^{-\alpha|j - i_0| - \beta|l - k_0|}\, a_0, \quad a_0 := a_{i_0, k_0} > 0.$$

where $\alpha$ characterizes the temporal decay rate and $\beta$ characterizes the spatial decay rate.

(A2) **Infinite temporal grid & finite spatial grid.** To conduct asymptotic analysis, we let $j \in \mathbb{Z}$ (temporal) but keep $l \in \{1, \ldots, s\}$ (spatial). Extending $j$ to $\mathbb{Z}$ only enlarges the sums we bound.

### Notation

$$Z := \sum_{j \in \mathbb{Z}} \sum_{l=1}^{s} a_{j,l}, \qquad Z_{\text{keep}} := \sum_{(j,l)\,:\,\tilde{M}_{i_0,j,k_0,l}=0} a_{j,l}, \qquad Z_{\text{out}} := Z - Z_{\text{keep}}.$$

Exact and masked softmax rows: $p_{j,l} = a_{j,l}/Z$, $\tilde{p}_{j,l} = a_{j,l}\mathbf{1}_{\{\tilde{M}=0\}}/Z_{\text{keep}}$. The total variation error can be calculated as follows by standard algebraic argument,

$$\left\| \tilde{p} - p \right\|_1 \;=\; 2\,\frac{Z_{\text{out}}}{Z}. \tag{1}$$

Because $a_0$ itself is in the sum, $Z \geq a_0$. Hence

$$\frac{Z_{\text{out}}}{Z} \;\leq\; \frac{Z_{\text{out}}}{a_0}. \tag{2}$$

**Mask geometry** For a temporal offset $\Delta t := |j - i_0| \geq 0$, define the bandwidth

$$w(\Delta t) \;:=\; \frac{s}{2^{\lfloor \log_2 \max(\Delta t, 1) \rfloor}} \;\in\; \{1, 2, 4, \ldots, s\}.$$

The mask keeps a spatial index $l$ iff $|l - k_0| \leq w(\Delta t)$ and the frame is one of the sub-sampled frames; otherwise $\tilde{M}_{i_0,j,k_0,l} = -\infty$.

Two kinds of approximation errors, therefore, appear:

(i) Spatial tails inside kept frames

For each $\Delta t$, the discarded spatial part satisfies

$$\sum_{d>w(\Delta t)} e^{-\beta d} \leq \frac{e^{-\beta(w(\Delta t)+1)}}{1-e^{-\beta}}.$$

Because $w(\Delta t) \geq \frac{s}{2}$ when $\Delta t \leq s$,

$$T_1 := 2C_{\text{rel}}a_0 \sum_{\Delta t \geq 0} e^{-\alpha\Delta t} \sum_{d>w(\Delta t)} e^{-\beta d} \leq \frac{4C_{\text{rel}}a_0}{(1-e^{-\alpha})(1-e^{-\beta})} e^{-\beta\left(\frac{s}{2}+1\right)}. \tag{3}$$

(ii) Frames skipped by the subsampling rule

For $\Delta t > s$, only every $K(\Delta t) = \left\lceil 2^{\lfloor \log_2 \Delta t \rfloor}/s \right\rceil$ frame is kept; the remainder contributes

$$T_2 := 2C_{\text{rel}}a_0 \frac{1+e^{-\beta}}{1-e^{-\beta}} \sum_{\Delta t > s} e^{-\alpha\Delta t} \leq \frac{2C_{\text{rel}}a_0\left(1+e^{-\beta}\right)}{(1-e^{-\beta})(1-e^{-\alpha})} e^{-\alpha(s+1)}. \tag{4}$$

**Total variation error**  Combine all equations above:

$$\left\|\tilde{p}-p\right\|_1 \leq C_{\text{rel}} \left[ \frac{8\,e^{-\beta\left(\frac{s}{2}+1\right)}}{(1-e^{-\alpha})(1-e^{-\beta})} + 4\frac{1+e^{-\beta}}{1-e^{-\beta}}\frac{e^{-\alpha(s+1)}}{1-e^{-\alpha}} \right] = O(C_{\text{rel}}e^{-\min\{\beta/2,\alpha\}s}).$$

This characterizes how the decay rates affect the approximation error.

## B  Additional Implementation Details

In terms of our LoRA fine-tuning for longer-video generation, we fine-tune HunyuanVideo [1] and Mochi 1 [22] at a global batch size of 1 with sequence parallelism, and train Wan 2.1 [7] with a global batch size of 8. All tuning experiments are conducted on 8 H100 GPUs. During training, we keep the first two DiT blocks with dense attention. Since there are 60, 48, and 40 blocks for HunyuanVideo, Mochi 1, and Wan2.1-14B, this only incurs negligible overhead. We train HunyuanVideo for 2× and 4× length video generation for 2400 and 1200 steps, respectively. We train Mochi 1 for 5000 steps for both 2× and 4× length video generation. We train Wan2.1-14B for 2500 steps for 2× length video generation. The LoRA rank is 128 for all training tasks.

## C  Visualization of the generated videos

In this section we compare Radial Attention against various baselines in video quality, and list our speedup in both training and inference.

### C.1  Default Video Length

We provide a visual comparison between the original dense attention, STA [47], and our Radial Attention on HunyuanVideo [1] and Wan2.1-14B [7]. We conduct experiments under 768p, 117 frames settings for HunyuanVideo, and 768p, 69 frames settings for Wan2.1-14B. As shown in Figure A and Figure B, Radial Attention has higher PSNR compared to STA [47], effectively maintaining the high fidelity of the original videos.

### C.2  Longer-video Length

We provide a visual comparison between the aforementioned baselines and Radial Attention on HunyuanVideo [1], Mochi 1 [22], and Wan2.1-14B [7]. We conduct experiments under the default resolution settings, which are 720p for HunyuanVideo and Wan2.1-14B, and 480p for Mochi 1. Moreover, we generate videos at 4× longer length for HunyuanVideo (21 seconds, 509 frames) and

Table A: We compare Radial Attention against another $\mathcal{O}(n \log n)$ attention baseline, Harmonic Series (HS). Radial Attention consistently outperforms it across all metrics.

| Model | Method | PSNR (↑) | SSIM (↑) | LPIPS (↓) | VisionReward (↑) |
|---|---|---|---|---|---|
| HunyuanVideo (117 frames) | HS | 27.0 | 0.881 | 0.119 | 0.136 |
| | **Ours** | **27.3** | **0.886** | **0.114** | **0.139** |

Table B: Ablation on the number of initial full-attention (warmup) steps for default-length video generation of Wan2.1-14B model. The 12-step warmup achieves the best performance across all metrics.

| Model | #Warmup Steps | PSNR (↑) | SSIM (↑) | LPIPS (↓) |
|---|---|---|---|---|
| Wan2.1-14B (69 frames) | 0 | 12.8 | 0.486 | 0.522 |
| | 4 | 18.5 | 0.693 | 0.267 |
| | 8 | 21.7 | 0.778 | 0.183 |
| | 11 | 23.2 | 0.813 | 0.151 |
| | **12 (Ours)** | **23.6** | **0.823** | **0.146** |
| | 13 | 23.5 | 0.819 | 0.150 |

Mochi 1(22 seconds, 667 frames), and 2× longer length for Wan2.1-14B (10 seconds, 161 frames). We use Vision Reward [69] to evaluate the generated videos. Figure C, Figure D, and Figure E demonstrate that Radial Attention achieves the highest average Vision Reward score compared to the baselines, well preserving the video quality even at longer-video settings.

### C.3 LoRA Compatibility Visual Results

As illustrated in Figure F, combining our extended-length LoRA with existing style LoRAs preserves visual quality while enabling longer-video generation. We observe that the content style generated by the merged LoRA exhibits subtle differences from the original LoRAs. This discrepancy is primarily attributed to the relatively small dataset used for training the extended-length LoRA, which may introduce a slight style bias that interacts with the style LoRA. We expect that training the length-extension LoRA on a more comprehensive dataset would help mitigate this issue.

## D  Ablations on Initial Dense-Attention Layers and Steps

We provide additional ablation studies to justify our design choices, including how many dense-attention timesteps and blocks we use to deliver the best video quality with the same computation budget, as well as the design of our $\mathcal{O}(n \log n)$ attention mask.

### D.1  Ablation on Initial Dense-Attention Steps

For default-length video generation, we follow SVG [8] to apply full attention to the first 25% of timesteps (12 steps) as a warmup for all methods. We ablate this setting in Table B. For fair comparison, we match the overall computation of all settings by adjusting the sparsity of our Radial Attention mask. The 12-step warmup achieves the best video quality across all metrics.

For 4× longer video generation, we apply full attention to the first 2 steps as a warmup during inference. The impact of different warmup steps on HunyuanVideo is shown in Table C. Computation is matched across all configurations. Using 2 warmup steps achieves the highest Vision Reward.

### D.2  Ablation on Initial Dense-Attention Layers

To better capture global information, we keep the first two layers as full attention during training. Table D reports results on HunyuanVideo when using 0, 1, 2, or 3 dense layers, under the same computation budget. Our choice of using 2 full-attention layers delivers the best video quality.

Table C: Ablation on the number of warmup steps for $4\times$ longer video generation. Two warmup steps yield the best Vision Reward.

| Model | #Warmup Steps | Vision Reward ($\uparrow$) |
|---|---|---|
| HunyuanVideo (117 frames) | 0 | 0.154 |
| | 1 | 0.160 |
| | **2 (Ours)** | **0.163** |
| | 3 | 0.157 |

Table D: Ablation on the number of initial full-attention (dense) layers during training. Using two full-attention layers yields the highest Vision Reward.

| Model | #Dense Layers | Vision Reward ($\uparrow$) |
|---|---|---|
| HunyuanVideo (117 frames) | 0 | 0.139 |
| | 1 | 0.156 |
| | **2 (Ours)** | **0.163** |
| | 3 | 0.157 |

### D.3 Comparison with Other $\mathcal{O}(n \log n)$ Sparsity Patterns

We additionally conduct an ablation study to validate the effectiveness of the sparsity pattern in our proposed $\mathcal{O}(n \log n)$ attention mask. Specifically, we compare Radial Attention with the Harmonic Series Decay Attention (HS), which features a computed diagonal width inversely proportional to its distance from the main diagonal. Table A presents quantitative results comparing Radial Attention with HS, demonstrating the superiority of Radial Attention.

## E    Broader Impacts

Radial Attention significantly reduces computational costs for video diffusion models, enabling longer-video generation with minimal fine-tuning efforts while maintaining quality. This paves the way for high-quality video creation tools for education and creative arts. Since Radial Attention accelerates self-attention to $\mathcal{O}(n \log n)$ complexity, it can accelerate video diffusion models and decrease energy consumption, leading to greener AI applications. This also helps the popularization of generative models. However, malicious users can misuse our method to create deepfakes and spread misinformation. The technology may also exacerbate the digital divide between those with and without access to the minimal necessary computational resources. To address these concerns, we advocate for responsible deployment, adherence to ethical standards, and the development of effective detection methods. We encourage the research community to continue advancing both efficient generation techniques and safeguards to ensure these powerful tools benefit society while minimizing potential harms. We will explicitly specify the usage permission of our code and models with proper licenses.

## F    License

Here, we show all the licenses for our used assets. Wan 2.1 [7], Mochi 1 [22], Diffusers, and STA [47] are under Apache-2.0 license. The license of HunyuanVideo [1] is here. SVG [8] and OpenVid-1M do not have an explicit license.

*Prompt: A shark is swimming in the ocean, featuring a steady and smooth perspective. Realistic, Natural lighting, Mysterious.*

Original HunyuanVideo
PFLOPs: 612 Latency: 1649s
Speedup: 1.0×

STA(FA3)
PSNR: 29.8
PFLOPs: 331 Latency: 719s
Speedup: 2.3×

Radial Attention (Ours)
PSNR: 31.2
PFLOPs: 339 Latency: 876s
Speedup: 1.9×

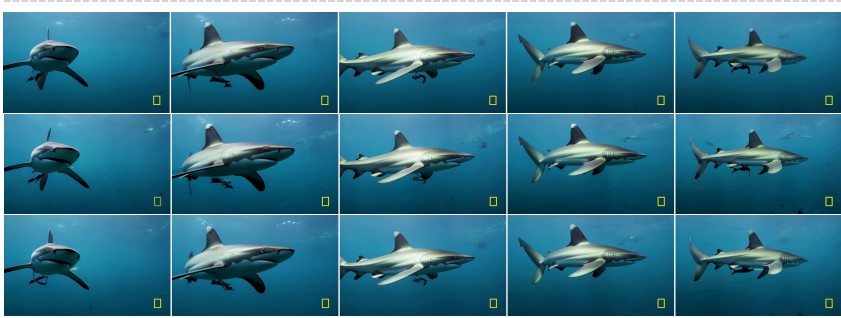

*Prompt: Martial artists exchanging fluid, powerful strikes in a serene, ancient temple courtyard, dust clouds rising in slow motion from every footfall and impact.*

Original HunyuanVideo
PFLOPs: 612 Latency: 1649s
Speedup: 1.0×

STA(FA3)
PSNR: 23.6
PFLOPs: 331 Latency: 719s
Speedup: 2.3×

Radial Attention (Ours)
PSNR: 26.1
PFLOPs: 339 Latency: 876s
Speedup: 1.9×

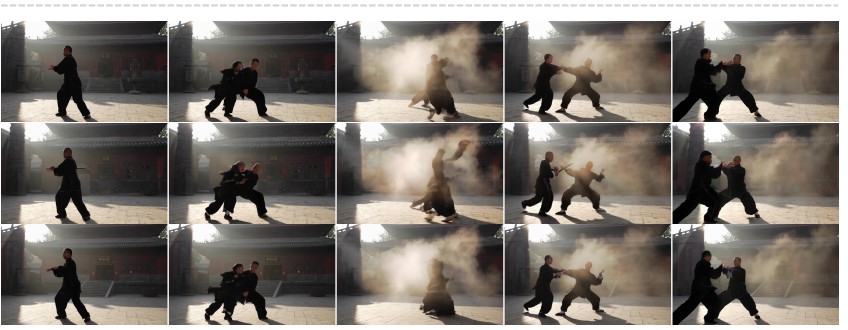

*Prompt: A couple in formal evening wear walks home and gets caught in a heavy downpour with umbrellas, surrealism style. Night lighting, Mysterious.*

Original HunyuanVideo
PFLOPs: 612 Latency: 1649s
Speedup: 1.0×

STA(FA3)
PSNR: 24.2
PFLOPs: 331 Latency: 719s
Speedup: 2.3×

Radial Attention (Ours)
PSNR: 25.5
PFLOPs: 339 Latency: 876s
Speedup: 1.9×

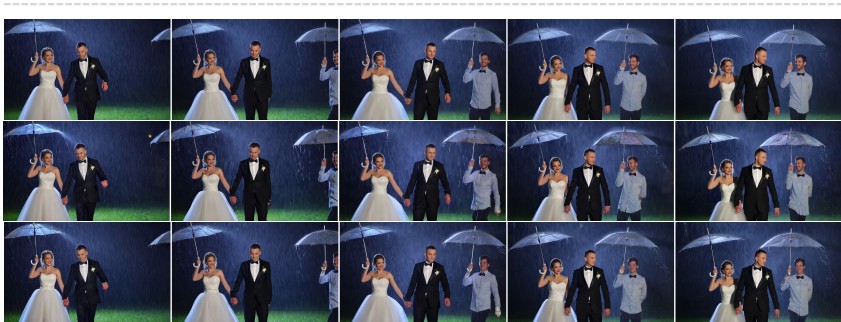

*Prompt: A dancer spinning with explosive energy under a sharp spotlight, loose fabric and fine dust swirling around her in a whirlwind of motion and emotion.*

Original HunyuanVideo
PFLOPs: 612 Latency: 1649s
Speedup: 1.0×

STA(FA3)
PSNR: 27.2
PFLOPs: 331 Latency: 719s
Speedup: 2.3×

Radial Attention (Ours)
PSNR: 28.8
PFLOPs: 339 Latency: 876s
Speedup: 1.9×

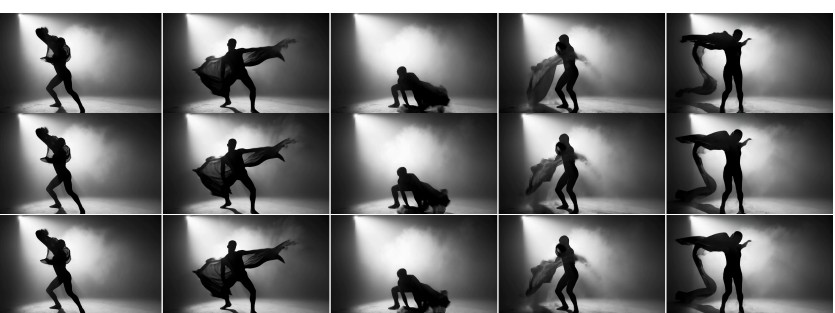

Figure A: Comparison of Dense Attention and Radial Attention on HunyuanVideo Text-to-Video generation at the default length (5 seconds, 117 frames, 768p).

*Prompt: A bus is stuck in traffic during rush hour. Realistic, Natural lighting, Tense.*

Original Wan2.1-14B
PFLOPs: 560 Latency: 1630s
Speedup: 1.0×

STA(FA3)
PSNR: 19.4
PFLOPs: 322 Latency: 812s
Speedup: 2.0×

Radial Attention (Ours)
PSNR: 22.2
PFLOPs: 323 Latency: 917s
Speedup: 1.8×

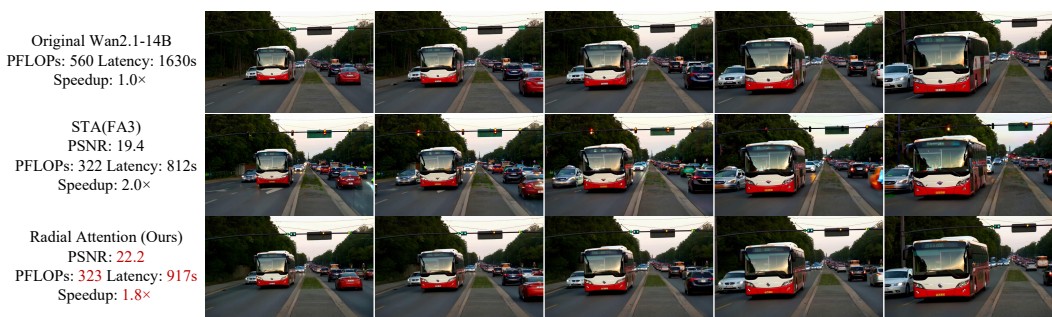

*Prompt: A solitary figure stands on a windswept cliff, their silhouette framed by a dramatic sunset, wearing a long, flowing coat that billows in the breeze.*

Original Wan2.1-14B
PFLOPs: 560 Latency: 1630s
Speedup: 1.0×

STA(FA3)
PSNR: 21.8
PFLOPs: 322 Latency: 812s
Speedup: 2.0×

Radial Attention (Ours)
PSNR: 23.6
PFLOPs: 323 Latency: 917s
Speedup: 1.8×

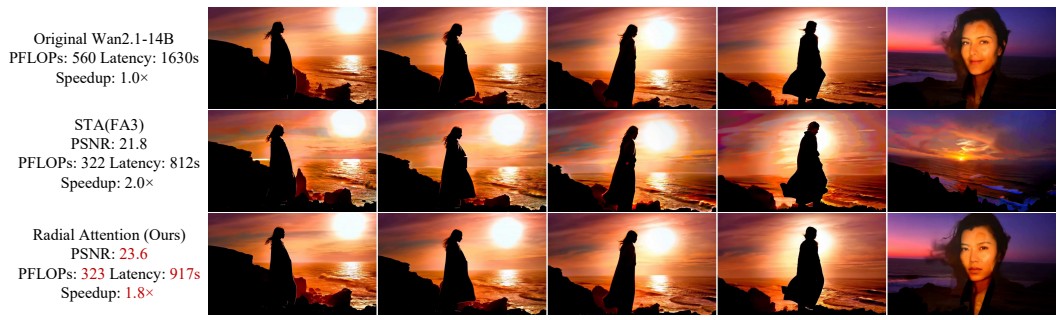

*Prompt: A breathtaking coastal beach in spring, with gentle waves lapping against the golden sand, is depicted in the vibrant, swirling brushstrokes of Van Gogh.*

Original Wan2.1-14B
PFLOPs: 560 Latency: 1630s
Speedup: 1.0×

STA(FA3)
PSNR: 19.6
PFLOPs: 322 Latency: 812s
Speedup: 2.0×

Radial Attention (Ours)
PSNR: 22.1
PFLOPs: 323 Latency: 917s
Speedup: 1.8×

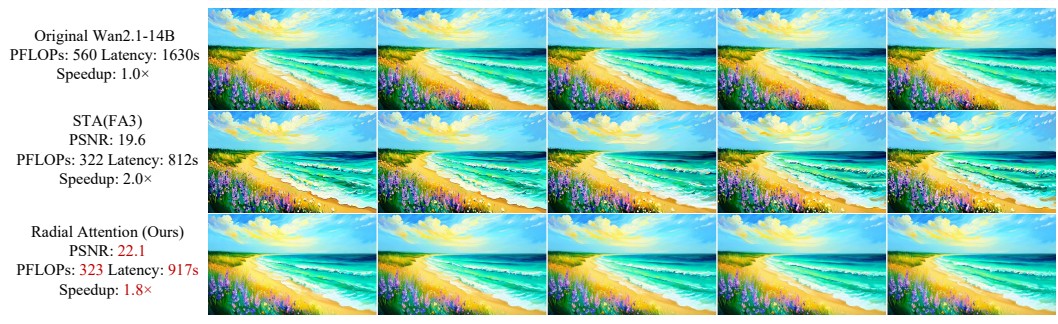

*Prompt: A teddy bear is swimming in the ocean. Realistic, Natural lighting, Mysterious*

Original Wan2.1-14B
PFLOPs: 560 Latency: 1630s
Speedup: 1.0×

STA(FA3)
PSNR: 18.5
PFLOPs: 322 Latency: 812s
Speedup: 2.0×

Radial Attention (Ours)
PSNR: 20.3
PFLOPs: 323 Latency: 917s
Speedup: 1.8×

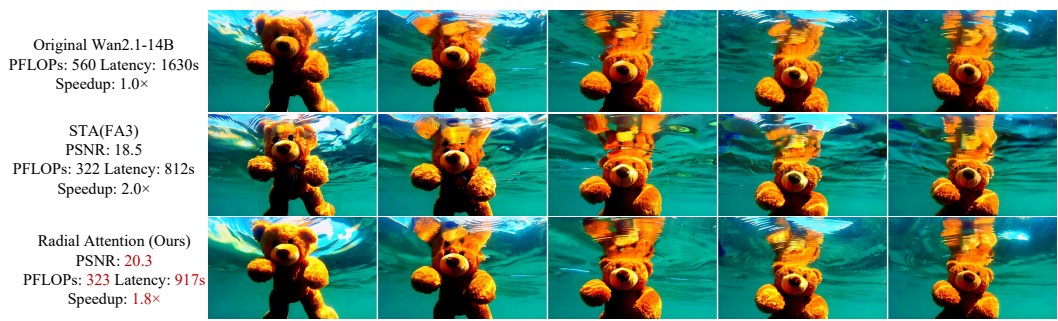

Figure B: Comparison of Dense Attention and Radial Attention on Wan2.1-14B Text-to-Video generation at the default length (4 seconds, 69 frames, 768p).

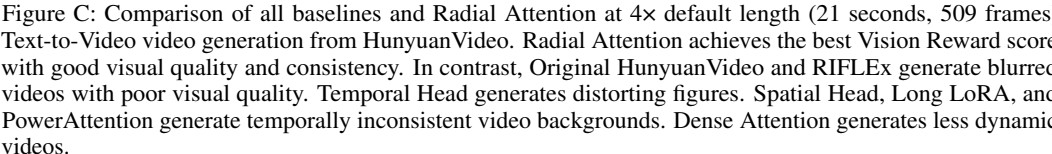

Figure C: Comparison of all baselines and Radial Attention at 4× default length (21 seconds, 509 frames) Text-to-Video video generation from HunyuanVideo. Radial Attention achieves the best Vision Reward score with good visual quality and consistency. In contrast, Original HunyuanVideo and RIFLEx generate blurred videos with poor visual quality. Temporal Head generates distorting figures. Spatial Head, Long LoRA, and PowerAttention generate temporally inconsistent video backgrounds. Dense Attention generates less dynamic videos.

Original Mochi 1, VisionReward: -0.041, Latency: 992s, Inference Speedup: 1.0×, Training Time: 0 GPU Hours

Mochi 1+LoRA+Spatial Head, VisionReward: 0.143, Latency: 382s, Inference Speedup: 2.6×, Training Time: 139 GPU Hours, Training Speedup: 2.8×

Mochi 1+LoRA+Temporal Head, VisionReward: -0.024, Latency: 393s, Inference Speedup: 2.5×, Training Time: 141 GPU Hours, Training Speedup: 2.8×

Mochi 1+LongLoRA, VisionReward: 0.037, Latency: 426s, Inference Speedup: 2.3×, Training Time: 152 GPU Hours, Training Speedup: 2.6×

Mochi 1+LoRA+PowerAttention, VisionReward: 0.090, Latency: 381s, Inference Speedup: 2.6×, Training Time: 138 GPU Hours, Training Speedup: 2.8×

Mochi 1+LoRA+Dense Attention, VisionReward: 0.130, Latency: 992s, Inference Speedup: 1.0×, Training Time: 394 GPU Hours, Training Speedup: 1.0×

Mochi 1+LoRA+Radial Attention (Ours), VisionReward: 0.182, Latency: 386s, Inference Speedup: 2.6×, Training Time: 139 GPU Hours, Training Speedup: 2.8×

*Prompt: A breathtaking coastal beach in the vibrant spring season, waves gently lap at the golden sandy shores. In black and white, the scene*
*captures the serene beauty of nature. A lone figure in a stylish beige windbreaker strolls along the edge of the water, casting occasional glances*
*towards the horizon. Seagulls fly overhead, their silhouettes stark against the clear blue sky. Soft dunes rise behind them, blending seamlessly*
*into the lush greenery of nearby trees.*

Original Mochi 1, VisionReward: -0.024, Latency: 992s, Inference Speedup: 1.0×, Training Time: 0 GPU Hours

Mochi 1+LoRA+Spatial Head, VisionReward: 0.046, Latency: 382s, Inference Speedup: 2.6×, Training Time: 139 GPU Hours, Training Speedup: 2.8×

Mochi 1+LoRA+Temporal Head, VisionReward: 0.008, Latency: 393s, Inference Speedup: 2.5×, Training Time: 141 GPU Hours, Training Speedup: 2.8×

Mochi 1+LongLoRA, VisionReward: 0.006, Latency: 426s, Inference Speedup: 2.3×, Training Time: 152 GPU Hours, Training Speedup: 2.6×

Mochi 1+LoRA+PowerAttention, VisionReward: 0.097, Latency: 381s, Inference Speedup: 2.6×, Training Time: 138 GPU Hours, Training Speedup: 2.8×

Mochi 1+LoRA+Dense Attention, VisionReward: 0.056, Latency: 992s, Inference Speedup: 1.0×, Training Time: 394 GPU Hours, Training Speedup: 1.0×

Mochi 1+LoRA+Radial Attention (Ours), VisionReward: 0.097, Latency: 386s, Inference Speedup: 2.6×, Training Time: 139 GPU Hours, Training Speedup: 2.8×

Figure D: Comparison of all baselines and Radial Attention at 4× default length (22 seconds, 667 frames) Text-to-Video video generation from Mochi 1. Radial Attention achieves the highest Vision Reward score because it has excellent visual quality and consistency. In contrast, Original Mochi 1 generates blurred videos with poor visual quality. Spatial Head, Temporal Head, Long LoRA, PowerAttention, and Dense Attention generate videos with either inconsistent backgrounds or inconsistent figures.

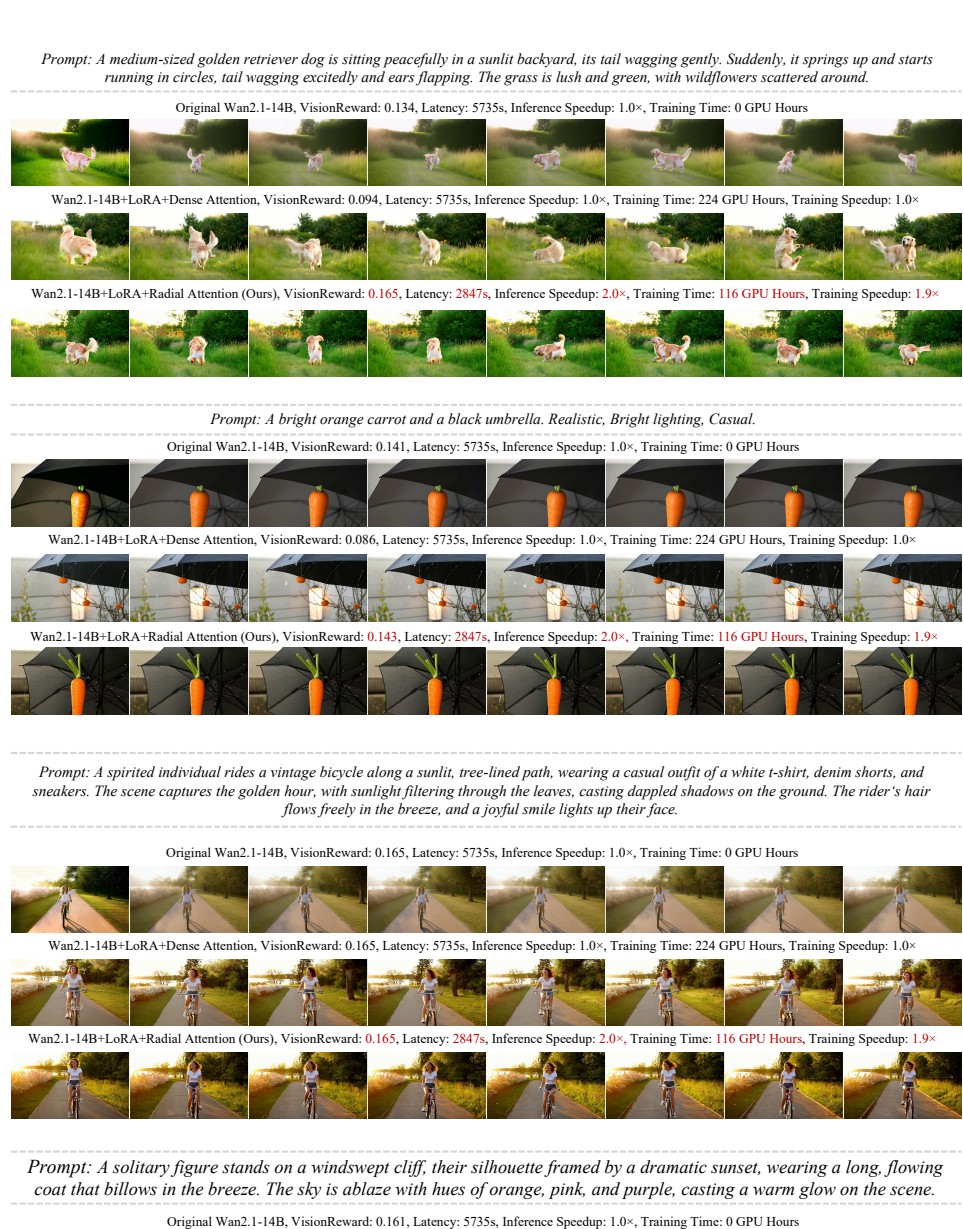

Figure E: Comparison of all baselines and Radial Attention at 2× default length (10 seconds, 161 frames) Text-to-Video video generation from Wan2.1-14B. Radial Attention achieves the highest Vision Reward score original Wan2.1-14B generates blurred videos and Dense Attention generates videos with inconsistent figures.

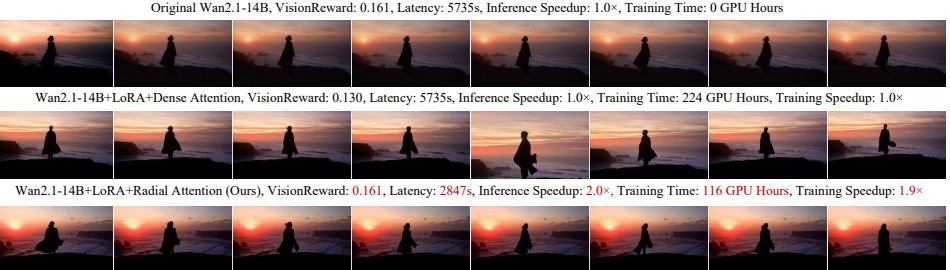

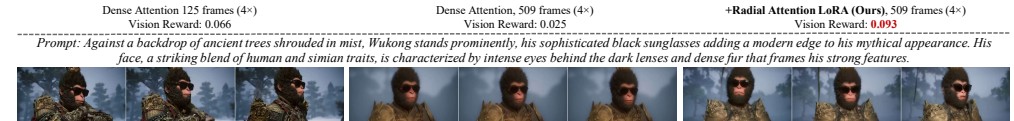

Figure F: Radial Attention LoRA is compatible with existing style LoRAs. On HunyuanVideo, it extends video length by 4× while maintaining a vision reward comparable to that of the original-length LoRA video.

