# OpenReview forum: "Radial Attention: $\mathcal{O}(n\log n)$ Sparse Attention with Energy Decay for Long Video Generation"
_NeurIPS.cc/2025/Conference — NeurIPS 2025 poster_

### Official Review · Reviewer_oEiS · 2025-06-20

**Clarity:** 3
**Significance:** 3
**Originality:** 3
**Rating:** 5
**Confidence:** 4

**Summary:**

This paper introduces Aura Attention, a novel sparse attention mechanism for video diffusion models that reduces computational complexity from O(n²) to O(n log n) by leveraging spatiotemporal energy decay—an observed phenomenon where attention scores decay exponentially with spatial/temporal distance between tokens. The method uses a static hardware-friendly mask that prioritizes local spatial interactions and exponentially shrinks the attention window with temporal distance. Evaluated on SOTA models (HunyuanVideo, Wan2.1, Mochi 1), Aura Attention achieves:

(1) 1.9× speedup at default video lengths with comparable quality.

(2) 4.4× fewer tuning costs and 3.7× faster inference for 4× longer videos when combined with LoRA fine-tuning.

The paper includes a theoretical analysis of complexity/error bounds, extensive experiments, and ablation studies.

**Questions:**

My main concerns are as follows:

1. The details of Fig.3 are vague. This is the key motivation of the paper, but how this figure is obtained is vague: (1) Are the results obtained on a single run or an average of a set of videos? (2) It is obtained on a single topic of content or diverse scenarios.

2. Does Fig.3 also hold for other models like Wan2.1 and Open-Sora?

3. The robustness of the observed Attention Spatiotemporal Energy Decay is unclear: does it still hold for more complex scenarios, e.g., large motion videos and videos with complex content?

4. Typo: Line 261, Line 273

**Ethical Concerns:**

["NO or VERY MINOR ethics concerns only"]

**Final Justification:**

The authors mostly address my concerns. The revision should be polished according to the comments in the rebuttal.

**Limitations:**

Yes.

**Quality:**

3

**Strengths And Weaknesses:**

Strengths:
+ Identifies and formalizes spatiotemporal energy decay (Fig. 3), grounding the approach in empirical analysis of attention maps. This provides a principled basis for sparsity.
+ Achieves O(n log n) complexity via a static mask (Fig. 4) with exponentially decaying compute density. This avoids costly dynamic profiling (e.g., SVG) and enables hardware optimization. Demonstrated 9× compute reduction for 509-frame videos (Fig. 5a) and 3.7× inference speedup on H100 GPUs.
+ Demonstrate acceleration on both inference and training with Lora, making potential applications in practice.

Weakness:
- The details of Fig.3 are vague. This is the key motivation of the paper, but how this figure is obtained is vague: (1) Are the results obtained on a single run or an average of a set of videos? (2) It is obtained on a single topic of content or diverse scenarios.
- The robustness of the observed Attention Spatiotemporal Energy Decay is unclear: does it still hold for more complex scenarios, e.g., large motion videos and videos with complex content?

---

> ### Author Rebuttal · Authors · 2025-07-31
>
> Thanks for your insightful and constructive reviews. Below, we provide point-by-point responses to each of your comments.
>
> ### How is Figure 3 Obtained?
>
> The attention scores shown in Figure 3 are averaged across all attention heads, layers, and timesteps, based on 10 diverse prompts from VBench [1]. These prompts cover a range of themes, including animals, sports, people, et cetera.
>
>
> ### Does Fig 3. also Hold for Wan2.1 14B or Open-Sora?
>
> Yes, both the attention decay phenomenon and our proposed exponential decay model hold for Wan2.1 14B [2] and Open-Sora [3], demonstrating that proposed "Spatiotemporal Decay" is universal for leading video generation models.
>
> For Wan2.1 14B [2], we conducted a quantitative regression analysis on the average attention using 10 diverse prompts from VBench [1]. Specifically, we fitted three functions to the distribution:
> - the inverse model $y = \frac{a}{(x-b)} + c$,
> - the inverse square model $y = \frac{a}{(x-b)^2} + c$
> - our proposed exponential decay model $y = \exp(-ax + b) + c$.
>
> The regression results, measured by $R^2$ (higher is better), is summarized in the table below. Our exponential decay model achieves the best fit for both spatial and temporal decay, with $R^2$ values up to 0.996.
>
> | Regression Model                   | Spatial Decay $R^2$ ($\uparrow$)    | Temporal Decay $R^2$ ($\uparrow$)   |
> |------------------------------------|-------------|--------------|
> | $y = \frac{a}{(x-b)} + c$          | 0.980       | 0.989        |
> | $y = \frac{a}{(x-b)^2} + c$        | 0.988       | 0.994        |
> | $y = \exp(-ax + b) + c$ **(Ours)** | **0.990**   | **0.996**    |
>
> We applied the same analysis to Open-Sora 2.0, and again our exponential model achieved the highest $R^2$ values—over 0.992 in both decays.
>
> | Regression Model                   | Spatial Decay $R^2$ ($\uparrow$)    | Temporal Decay $R^2$ ($\uparrow$)   |
> |------------------------------------|-------------|--------------|
> | $y = \frac{a}{(x-b)} + c$          | 0.986       | 0.996        |
> | $y = \frac{a}{(x-b)^2} + c$        | 0.988       | 0.996        |
> | $y = \exp(-ax + b) + c$ **(Ours)** | **0.992**   | **0.997**    |
>
> These results demonstrate that both models follow the exponential Spatialtemporal Energy Decay.
>
> ### Robustness of Spatiotemporal Energy Decay in Complex Scenarios
>
> As requested by the reviewer, we validate our exponential decay model in the complex scenarios. We collected 10 high-motion prompts (e.g. dancing, competitive sports, and culinary action) with detailed and complex descriptions. Here we provide an example prompt.
>
> > breakdancer caught mid-freeze, balancing on one hand while the crowd forms a vibrant, cheering circle, energy radiating through low ambient street lighting.
>
> Similarly, we average all the attention maps of Wan2.1 14B [2] on these prompts and then perform the regression analysis on the attention scores. The results are reported below. Our exponential decay model remains the best fit, achieving up to 0.997 $R^2$, demonstrating that the decay pattern holds in complex scenarios.
>
> | Regression Model                   | Spatial Decay $R^2$ ($\uparrow$)    | Temporal Decay $R^2$ ($\uparrow$)   |
> |------------------------------------|-------------|--------------|
> | $y = \frac{a}{(x-b)} + c$          | 0.976       | 0.987        |
> | $y = \frac{a}{(x-b)^2} + c$        | 0.987       | 0.993        |
> | $y = \exp(-ax + b) + c$ **(Ours)** | **0.989**   | **0.997**    |
>
> We also evaluated the visual quality of Aura Attention on these complex prompts for default-length generation. As shown in below table, the similarity metrics demonstrate that our approach maintains high video fidelity on both VBench prompts used in our paper and the complex prompts, indicating robustness of our method.
>
> |                     | PSNR ($\uparrow$) | SSIM ($\uparrow$) | LPIPS ($\downarrow$) |
> |---------------------|-------------------|--------------------|----------------------|
> | VBench Prompts            | **23.6**              | 0.823              | **0.146**                |
> | Complex Prompts  | 23.1              | **0.842**              | 0.178                |
>
> We also compare Aura Attention against the full attention for 4$\times$ longer video generation of HunyuanVideo after LoRA tuning. It is 4.4× faster in training and 3.7× faster in inference, while also delivering a slightly higher Vision Reward score:
>
> | Method                  | Vision Reward |
> |-------------------------|--------------------|
> | Full Attention | 0.102             |
> | **Our Aura Attention**  | **0.105**             |
>
> ### Typos
> Thanks for pointing these out! We will correct them in our manuscript.
>
> [1] VBench: Comprehensive Benchmark Suite for Video Generative Models
>
> [2] Wan: Open and Advanced Large-Scale Video Generative Models
>
> [3] Open-Sora 2.0: Training a Commercial-Level Video Generation Model in \$200k

---

> > ### Comment · Reviewer_oEiS · 2025-08-04
> >
> > Thanks for the response from the authors.
> > I suggest the authors to further polish the revision following the rebuttal once accepted.
> > I raised my score accordingly.

---

### Official Review · Reviewer_aPz8 · 2025-06-29

**Clarity:** 3
**Significance:** 3
**Originality:** 4
**Rating:** 5
**Confidence:** 4

**Summary:**

This paper introduces Aura Attention, an static attention mask with O(n log n) complexity wrt. number of frames suitable for long video generation. The design of this attention mask is motivated by empirical observations on the attention patterns in HunyuanVideo. Long video generaiton is enabled through LoRA finetuning, since the sparse attention mask reduces the prohibitive cost of training on long videos. Experiments are done on three backbones at the default video length and extended length, showing that Aura Attention speeds up both training and inference while achieving good quality compared to baselines.

**Questions:**

- Figure 5 shows that Aura Attention enables better video quality for long video generation finetuning. Can the authors give some conceptual or theoretical analysis behind this?

**Ethical Concerns:**

["NO or VERY MINOR ethics concerns only"]

**Final Justification:**

My concerns are addressed in the rebuttal. I maintain my rating of accept, since the paper introduces a novel attention pattern for video generation, demonstrate strong experiment results, and should have a high impact in the field.

**Limitations:**

Yes.

**Quality:**

3

**Strengths And Weaknesses:**

## Strengths
- Good writing quality.
- Strong experiment results on long video generation in tab. 2. A wide range of baselines are adopted in tab. 1, 2.
- The design of Aura Attention mask is intuitive, innovative, and motivated through empirical observations shown in fig. 3.
- Complexity and error bound for Aura Attention are provided.
- Aura Attention enables long video generation finetuning with modest training cost, which is a core challenge for long video generation.

## Weaknesses
- The complexity of Aura Attention, while $\mathcal{O}(n \log{n})$ wrt. number of frames, still have $\mathcal{O}(n^2)$ wrt. the spatial resolution, which is shown in Eq. 6 and discussed in Sec. 6 as a limitation. This limits the main application of Aura Attention to long video generation, while existing methods (STA, SVG) are either more efficient or have better quality in the short video setting, as shown in Tab. 1.
- No ablation studies are done. Possible ablation studies include the usage of attention sink, the width of the central band, the width scale between consecutive bands (currently 2x), whether to reduce the frequency of diagonals if the diagonal width falls below 1, etc. Conducting some ablation studies can provide insights into the design space of Aura Attention.

---

> ### Author Rebuttal · Authors · 2025-07-31
>
> Thanks for your insightful and constructive reviews. Below, we provide point-by-point responses to each of your comments.
>
> ### Complexity with Respect to Spatial Resolution
>
> While Aura Attention has a theoretical complexity of $\mathcal{O}(n^2)$ with respect to spatial resolution, its practical speedup does not degrade as resolution increases for a fixed video length. As discussed in Section 4.2, our mask design guarantees constant sparsity across resolutions for a fixed number of frames. The table below compares the per-step latency of Wan2.1 14B using original full attention versus our Aura Attention for generating 5-second videos. The speedup becomes more significant at higher resolutions, since attention computation scales quadratically with the number of pixels per frame, while most other operations scale linearly.
>
> | Resolution      | Attention Type       | Single-Step Latency (s)     | Speedup |
> |-----------------|----------------------|------------------|--|
> | 720p (1280×720) | Full        | 32.9           | -- |
> |                 | **Our Aura**| **15.4**  | **2.1×** |
> | 1080p (1920×1080)| Full       | 136.6          | -- |
> |                 | **Our Aura**| **53.1**   | **2.6×** |
> | 2K (2160×1440)  | Full        | 437.8          | -- |
> |                 | **Our Aura**| **127.5** | **3.4×** |
>
> Moreover, recent advances in video VAE have achieved remarkable progress in spatial compression, pushing the compression ratio from $8\times8$ to $16\times16$ [2] or even $32\times32$ [3]. These methods can be combined with Aura Attention to mitigate the quadratic cost of spatial resolution.
>
> ### Short Video Generation
>
> We would like to clarify that in Table 1 Aura Attention achieves video quality comparable to SVG [1] and outperforms STA [4] on Hunyuan Video under the same computation budget. On Wan2.1 14B, Aura Attention beats both baselines in quality. Regarding latency, Aura Attention matches SVG on both models. STA uses FlashAttention-3 (FA3) [5] as its backend, providing slightly higher speedup. Aura Attention can reach similar performance by switching the backend from FA2 to FA3, achieving a speedup of 2.27× on Hunyuan Video and 1.98× on Wan2.1 as shown in the below table, on par with STA.
>
> | Model         | Method         | PSNR (↑) | SSIM (↑) | LPIPS (↓) | Vision Reward (↑) | Latency (s) | Speedup |
> |---------------|----------------|----------|----------|-----------|--------------------|--------------|---------|
> | Hunyuan Video | STA (FA3)      | 26.7     | 0.866    | 0.167     | 0.132              | **719**          | **2.29×**   |
> |               | SVG (FA2)      | 27.2     |**0.895**| **0.114**     | **0.144**              | 867          | 1.90×   |
> |               | **Ours (FA2)** | **27.3** |   0.886| **0.114** | 0.139          | 876      | 1.88× |
> |               | **Ours (FA3)** | **27.3** |   0.886| **0.114** | 0.139          | 724      | 2.27× |
> | Wan2.1-14B    | STA (FA3)      | 22.9     | 0.830    | 0.171     | **0.132**              | **812**          | **2.01×**   |
> |               | SVG (FA2)           | 23.2     | 0.825    | 0.202     | 0.114              | 949          | 1.71×   |
> |               | **Ours (FA2)**       | **23.9** | **0.842**| **0.163** | 0.128          | 917      | 1.77× |
> |               | **Ours (FA3)**       | **23.9** | **0.842**| **0.163** | 0.128          | 825      | 1.98× |
>
>
> ### Ablation Studies
>
> As requested by the reviewer, we provide additional ablation studies to justify our design choices. The experiments were conducted on Wan2.1 14B at a resolution of $1280\times768$ with 117 frames. We will include them in our manuscript.
>
> **Attention Sink**
>
> We follow SVG [1] to preserve attention to the first frame (attention sink), which retains a significant portion of the attention energy. Quality metrics listed in the table below show that removing the attention sink could result in noticeable degradation of video quality. Here we align the mask sparsity with and without the attention sink for fair comparison.
>
> | Configuration                | PSNR ($\uparrow$)  | SSIM ($\uparrow$)  | LPIPS ($\downarrow$) |
> |------------------------------|-------|-------|-------|
> | Ours w/o Sink       | 21.1  | 0.767 | 0.209 |
> | **Ours**                 | **23.6**  | **0.823** | **0.146** |
>
> **Width of Central Band**
>
> The table below shows results of different central band widths. A central band width of $d$ means attention between frames $i$ and $j$ where $|i - j| \leq d$. For fair comparison, we align mask sparsity across all settings. Narrower bands (e.g., $d=0$) are suboptimal as they miss some important interactions between neighboring frames, which is crucial for video quality, while wider bands (e.g., $d=2$) spend too much computation budget on the central areas, failing to capture the critical tokens in the distant frames.
>
> | Central Band Width | PSNR ($\uparrow$)  | SSIM ($\uparrow$)  | LPIPS ($\downarrow$) |
> |------------------------------|-------|-------|-------|
> | 0     | 23.5  | 0.810 | 0.148 |
> | **1 (Ours)** | **23.6**  | **0.823** | **0.146** |
> | 2    | 18.8  | 0.660 | 0.275 |
>
>
> **Width Scale Between Consecutive Bands**
>
> The table below shows results of different width width scale between consecutive bands. For fair comparison, we also keep mask sparsity consistent across all settings. Our width scale of 2 achieves the best video quality. Smaller scales (e.g., 1.5) reduce attention on the central band, while larger scales (e.g., 3 or 4) limit computation on important tokens in distant frames, making them suboptimal.
>
> | Decay Rate | PSNR(↑) | SSIM(↑) | LPIPS(↓) |
> |------------|---------|---------|----------|
> | 1.5| 23.2    | 0.813   | 0.154    |
> | **2 (Ours)** | **23.6**| **0.823**| **0.146**|
> | 3  | 23.4    | 0.819   | 0.149    |
> | 4  | 21.8    | 0.759   | 0.182    |
>
> **Whether to Reduce the Diagonal Frequency when the Width is below 1**
>
> We reduce the diagonal frequency when the width is below 1 to maintain $\mathcal{O}(n\log n)$ complexity. Otherwise, long videos would incur quadratic computation cost with respect to the number of frames, despite potential quality gains. Therefore, this ablation is not feasible in our design choice. We appreciate your understanding and are open to further discussion on this.
>
> ### Why Aura Attention is Better for Longer Video Generation
>
> For long video generation, the large number of tokens makes dense attention inefficient and hard to focus on key interactions. Aura Attention addresses this by leveraging the Spatiotemporal Energy Decay to preserve only important interactions, improving both learning efficiency and performance.
>
> [1] Sparse VideoGen: Accelerating Video Diffusion Transformers with Spatial-Temporal Sparsity, ICML 2025
>
> [2] Wan 2.2: Leading AI Video Generation Model
>
> [3] Open-Sora 2.0: Training a Commercial-Level Video Generation Model in \$200k
>
> [4] Fast Video Generation with Sliding Tile Attention, ICML 2025
>
> [5] FlashAttention-3: Fast and Accurate Attention with Asynchrony and Low-precision

---

> > ### Author Response · Authors · 2025-08-06
> >
> > Dear Reviewer aPz8,
> >
> > As the deadline of the discussion approaches, we wonder if you could kindly consider updating the score if our responses have addressed your concerns. If you have any remaining questions or need further clarification, we would be happy to discuss them with you.
> >
> > Thank you once again for your valuable valuable efforts in reviewing our paper.
> >
> > Best regards, The Authors

---

> > > ### Comment · Reviewer_aPz8 · 2025-08-08
> > >
> > > Thanks for the detailed rebuttal and new experiments. My questions are addressed. The additional ablation validates the default design. I have no further questions. I am maintaining my score.

---

### Official Review · Reviewer_XyA8 · 2025-07-03

**Clarity:** 3
**Significance:** 3
**Originality:** 3
**Rating:** 4
**Confidence:** 4

**Summary:**

This paper introduces Aura Attention, a sparse attention mechanism designed for efficient long video generation in diffusion models. The key innovation is inspired by the observation of spatiotemporal energy decay in attention scores, where post-softmax scores decrease with increasing spatial and temporal distance between tokens. Aura Attention translates this decay into exponentially decaying compute density, achieving O(n log n) computational complexity. The method uses a static attention mask that focuses on spatially nearby tokens with a shrinking window size as temporal distance increases. Experiments on models like HunyuanVideo and Wan2.1-14B show up to 1.9× speedup at default lengths and 3.7× speedup for 4× longer videos, while reducing training costs by 4.4× with minimal fine-tuning using LoRA. The approach maintains video quality comparable to full attention baselines.

**Questions:**

Please refer to the weakness part.

**Ethical Concerns:**

["NO or VERY MINOR ethics concerns only"]

**Final Justification:**

Keep my original rating.

**Limitations:**

Please refer to the weakness part.

**Quality:**

2

**Strengths And Weaknesses:**

Strength：
- The integration of physical energy decay principles into attention mechanism design is innovative, providing a principled way to reduce computation while preserving critical dependencies.
- Achieves O(n log n) complexity, addressing the quadratic scaling issue of full attention, which is crucial for long video generation.
Empirical Validation: Comprehensive experiments across multiple large video models (HunyuanVideo, Wan2.1-14B) demonstrate consistent speedups and quality preservation.
- Enables longer video generation (up to 4× extension) with minimal fine-tuning, reducing both training costs and inference latency significantly.

Weakness：
- The complexity still scales quadratically with spatial resolution, which may limit performance for very high-resolution videos.
- The model relies on a simplified exponential decay function for attention scores, which may not fully capture complex spatiotemporal dependencies in real-world videos.
- The static mask design might struggle with dynamic scenes requiring long-range temporal dependencies beyond the predefined decay  pattern.
- Some baselines (e.g., SANA) are not fully optimized, and comparisons may not reflect the latest advancements in linear attention. Discussing more linear attention-based image generation methods here would be better.

---

> ### Author Rebuttal · Authors · 2025-07-31
>
> Thanks for your insightful and constructive reviews. Below, we provide point-by-point responses to each of your comments.
>
> ### Complexity with Respect to Spatial Resolution
>
> While Aura Attention has a theoretical complexity of $\mathcal{O}(n^2)$ with respect to spatial resolution, its practical speedup does not degrade as resolution increases for a fixed video length. As discussed in Section 4.2, our mask design guarantees constant sparsity across resolutions for a fixed number of frames. The table below compares the per-step latency of Wan2.1 14B using original full attention versus our Aura Attention for generating 5-second videos. The speedup becomes more significant at higher resolutions, since attention computation scales quadratically with the number of pixels per frame, while most other operations scale linearly.
>
> | Resolution      | Attention Type       | Single-Step Latency (s)     | Speedup |
> |-----------------|----------------------|------------------|--|
> | 720p (1280×720) | Full        | 32.9           | -- |
> |                 | **Our Aura**| **15.4**  | **2.1×** |
> | 1080p (1920×1080)| Full       | 136.6          | -- |
> |                 | **Our Aura**| **53.1**   | **2.6×** |
> | 2K (2160×1440)  | Full        | 437.8          | -- |
> |                 | **Our Aura**| **127.5** | **3.4×** |
>
> Moreover, recent advances in video VAE have achieved remarkable progress in spatial compression, pushing the compression ratio from $8\times8$ to $16\times16$ [2] or even $32\times32$ [3]. These methods can be combined with Aura Attention to mitigate the quadratic cost of spatial resolution.
>
> ### Robustness to Complex Spatiotemporal Dependencies
>
> To evaluate whether our exponential decay model can capture complex spatiotemporal dependencies, we sampled 10 prompts from the LongVILA [4] supervised fine-tuning dataset. These prompts describe complex real-world scenes that demand high spatiotemporal dependencies due to intricate interactions and dynamic changes over time. A sample prompt is displayed below.
> > The woman shows the stickers to the camera, talks about them, places them on the table, and at one point sticks a sticker on her hand.
>
> We then conducted a quantitative regression analysis on the average attention score distribution of Wan2.1 14B on these prompts. Specifically, we fit three models to the spatial and temporal attention decay curves:
> - inverse model $y = \frac{a}{(x-b)} + c$
> - inverse square model $y = \frac{a}{(x-b)^2} + c$
> - our proposed exponential decay model $y = \exp(-ax + b) + c$.
>
> The regression results, measured by $R^2$ (higher is better), is summarized in the table below. The attention score distribution nicely fit our exponential decay model, achiving $R^2$ up to 0.992. This means even for complex real-world scenes, our model still holds. See the following response for the video quality.
>
> | Regression Model                   | Spatial Decay $R^2$ ($\uparrow$)    | Temporal Decay $R^2$ ($\uparrow$)   |
> |------------------------------------|-------------|--------------|
> | $y = \frac{a}{(x-b)} + c$          | 0.980       | 0.987        |
> | $y = \frac{a}{(x-b)^2} + c$        | 0.989       | 0.991        |
> | $y = \exp(-ax + b) + c$ **(Ours)** | **0.989**   | **0.992**    |
>
>
> ### Capability of Capturing Long-Range Dependencies
>
> As mentioned in Section 5.1 in our paper, we preserve full attention in the first two layers during both training and inference, and apply a 2-step full-attention warmup during inference. This setup helps the model effectively capture global information. Additionally, stacking multiple Aura Attention layers expands the receptive field, allowing information to propagate across distant tokens. Therefore, our method is capable of capturing long-range dependencies even in dynamic scenes.
>
> To support this, we further evaluate default-length video generation on Wan2.1 14B using the aforementioned complex prompts from LongVILA [4]. As shown in below table, the similarity metrics demonstrate that our approach maintains high video fidelity on both VBench prompts used in our paper and the LongVILA prompts, indicating robust performance across different scenarios.
>
> |Prompt Set| PSNR($\uparrow$) | SSIM($\uparrow$) | LPIPS($\downarrow$) |
> |----------------------|------------------|------------------|---------------------|
> | VBench Prompts            | 23.6             | 0.823            | 0.146               |
> | LongVILA Prompts   | 23.0             | 0.823            | 0.151               |
>
> We also report the video quality of 4$\times$ longer video generation on HunyuanVideo using LongVILA prompts in the below table. Aura Attention matches the Vision Reward of full attention, demonstrating strong long-range modeling capabilities.
>
> | Method                  | Vision Reward ($\uparrow$)|
> |-------------------------|--------------------|
> | Full Attention  | 0.128             |
> | **Aura Attention (Ours)** | **0.128** |
>
> ### Additional Linear Attention Baselines
>
> As suggested by the reviewer, we adapted two more image-based linear attention methods—LinFusion [5] and Focused Linear Attention [6]—for video generation. The table below shows their performance on 4× longer video generation after LoRA fine-tuning on HunyuanVideo, measured by Vision Reward. As mentioned in our paper, due to the substantial architectual discrepency between the linear attention and the original softmax attention, these baselines fail to the learn the weight distributions during the fine-tuning.
>
>
> | Method          | Vision Reward ($\uparrow$) |
> |-----------------|--------------------|
> | LinFusion       | -0.209             |
> | Focused Linear Attention  | -0.205             |
> | **Aura Attention (Ours)**|**0.134**        |
>
> [1] Sparse VideoGen: Accelerating Video Diffusion Transformers with Spatial-Temporal Sparsity, ICML 2025
>
> [2] Wan 2.2: Leading AI Video Generation Model
>
> [3] Open-Sora 2.0: Training a Commercial-Level Video Generation Model in \$200k
>
> [4] LongVILA: Scaling Long-Context Visual Language Models for Long Videos, ICLR 2025
>
> [5] LinFusion: 1 GPU, 1 Minute, 16K Image
>
> [6] FLatten Transformer: Vision Transformer using Focused Linear Attention, ICCV 2023

---

> > ### Comment · Reviewer_XyA8 · 2025-08-03
> >
> > Thanks for the author's rebuttal. I keep my rating.

---

### Official Review · Reviewer_E5WB · 2025-07-12

**Clarity:** 3
**Significance:** 3
**Originality:** 2
**Rating:** 4
**Confidence:** 4

**Summary:**

This paper introduces Aura Attention, a sparse attention mechanism designed to mitigate the quadratic computational cost of generating long videos with diffusion models. The central motivation is a phenomenon the authors call "Spatiotemporal Energy Decay," where attention scores are observed to diminish with increasing distance between tokens in both space and time. Aura Attention implements a static attention mask that mimics this decay by exponentially shrinking the attention window as temporal distance grows, achieving a reported complexity of O(nlogn). This allows pre-trained models to be adapted for longer video generation through efficient LoRA fine-tuning. The authors test their method on several large video models (Hunyuan Video, Wan2.1-14B, Mochi 1) and report substantial improvements in inference speed and training cost while preserving video quality.

**Questions:**

Please read the weakness of the manuscript provided above.

**Ethical Concerns:**

["NO or VERY MINOR ethics concerns only"]

**Limitations:**

Yes.

**Paper Formatting Concerns:**

None.

**Quality:**

2

**Strengths And Weaknesses:**

Strengths:
1.	The paper's primary strength lies in its empirical results. The method delivers significant, practical speedups (e.g., up to 3.7x in inference for 4x longer videos) while maintaining video quality across multiple SOTA models. This is a non-trivial engineering achievement.
2.	The concept of "Spatiotemporal Energy Decay" is an intuitive and appealing motivation for designing a sparsity pattern. It provides a clear, high-level story for why the proposed attention mask might be effective.

Weakness:
1.	The paper’s scientific foundation rests on the "Spatiotemporal Energy Decay" phenomenon, which is modeled as an exponential decay. While this is a compelling narrative, the evidence provided is primarily qualitative, based on visual inspection of the plots in Figure3. For a method presented as "principled," the lack of a rigorous quantitative analysis to validate this core assumption is a significant weakness. This makes the approach feel more like a well-motivated heuristic than a scientifically-validated model.
2.	The paper omits critical ablation studies that would justify its specific implementation. Key decisions feel arbitrary and are not supported by experiments: The model retains full attention for the first few layers and initial denoising timesteps, which is noted as being "crucial for video quality". However, there is no analysis of how this parameter was chosen or its impact on the performance/speed trade-off.

---

> ### Author Rebuttal · Authors · 2025-07-31
>
> Thanks for your insightful and constructive reviews. Below, we provide point-by-point responses to each of your comments.
>
> ### Quantitative Analysis on "Spatiotemporal Energy Decay"
> To validate the exponential decay modeling of our proposed "Spatiotemporal Energy Decay" phenomenon, we conducted a quantitative regression analysis on the average attention score distribution of Wan2.1 14B using 10 diverse prompts from VBench [1]. Specifically, we fit three functions to the distribution:
> - the inverse model $y = \frac{a}{(x-b)} + c$,
> - the inverse square model $y = \frac{a}{(x-b)^2} + c$,
> - our proposed exponential decay model $y = \exp(-ax + b) + c$.
>
> The regression performance, measured by $R^2$ (higher is better), is summarized in the table below. Our exponential model achieves the highest $R^2$ values in both spatial decay and temporal decay, up to 0.996. This indicates that our proposed decay modeling is an excellent fit to the data.
>
> | Regression Model                   | Spatial Decay $R^2$ ($\uparrow$)    | Temporal Decay $R^2$ ($\uparrow$)   |
> |------------------------------------|-------------|--------------|
> | $y = \frac{a}{(x-b)} + c$          | 0.980       | 0.989        |
> | $y = \frac{a}{(x-b)^2} + c$        | 0.988       | 0.994        |
> | $y = \exp(-ax + b) + c$ **(Ours)** | **0.990**   | **0.996**    |
>
> ### Ablations on Initial Full-Attention Layers and Steps
> As requested by the reviewer, we provide additional ablation studies to justify our design choices. We will include them in our manuscript.
>
> For default-length video generation, we follow SVG [2] to apply full attention to the first 25% of timesteps (12 steps) as a warmup for all methods. We ablate this setting in the table below. For fair comparison, we match the overall computation of all settings by adjusting the sparsity of our Aura Attention mask. The 12-step warmup achieves the best video quality across all metrics.
>
> | #Warmup Steps                | PSNR ($\uparrow$)  | SSIM ($\uparrow$) | LPIPS ($\downarrow$) |
> |-----------------------------|-------|-------|-------|
> | 0                           | 12.8  | 0.486 | 0.522 |
> | 4                           | 18.5  | 0.693 | 0.267 |
> | 8                           | 21.7  | 0.778 | 0.183 |
> | 11                          | 23.2  | 0.813 | 0.151 |
> | **12 (Ours)**               | **23.6**| **0.823**| **0.146**|
> | 13                          | 23.5  | 0.819 | 0.150 |
>
>
> For 4× longer video generation, we apply full attention to the first 2 steps as a warmup during inference. The table below shows the impact of different warmup steps on HunyuanVideo. Similarly, we match computation of all configurations. Using 2 warmup steps achieves the highest Video Reward.
>
> | #Warmup Steps | Vision Reward ($\uparrow$) |
> |---------------|----------------------------|
> | 0             | 0.154                      |
> | 1             | 0.160                      |
> | 2 (**Ours**)  | **0.163**                 |
> | 3             | 0.157                      |
>
> To better capture global information, we keep the first two layers as full attention during training. The table below reports results on HunyuanVideo when using 0, 1, 2, or 3 dense layers, under the same computation budget. Our choice of using 2 full-attention layers delivers the best video quality.
>
>
> | #Dense Layers | Vision Reward ($\uparrow$) |
> |---------------|----------------------------|
> | 0             | 0.139                      |
> | 1             | 0.156                      |
> | 2 (**Ours**)  | **0.163**                      |
> | 3             | 0.157                      |
>
>
> [1] VBench: Comprehensive Benchmark Suite for Video Generative Models, CVPR 2024
>
> [2] Sparse VideoGen: Accelerating Video Diffusion Transformers with Spatial-Temporal Sparsity, ICML 2025

---

> > ### Author Response · Authors · 2025-08-06
> >
> > Dear Reviewer E5WB,
> >
> > As the deadline of the discussion approaches, we wonder if you could kindly consider updating the score if our responses have addressed your concerns. If you have any remaining questions or need further clarification, we would be happy to discuss them with you.
> >
> > Thank you once again for your valuable valuable efforts in reviewing our paper.
> >
> > Best regards, The Authors

---

> > ### Author Response · Authors · 2025-08-08
> >
> > Dear Reviewer E5WB,
> >
> > As we approach the final day of the discussion phase, we would be grateful if you could consider updating the score if our responses have satisfactorily addressed your concerns. Please feel free to reach out if you have any further questions or need additional clarifications, as we are more than happy to continue the discussion with you.
> >
> > Thank you once again for your valuable valuable efforts in reviewing our paper.
> >
> > Best regards, The Authors

---

### Decision · Program_Chairs · 2025-09-17

**Decision:**

Accept (poster)

**Comment:**

Summary:

This paper introduces Aura Attention, a novel sparse attention mechanism for video diffusion models, motivated by the empirical observation of "Spatiotemporal Energy Decay" - the tendency for attention scores to decay exponentially with increasing spatial and temporal distance between tokens. The proposed static attention mask achieves O(n log n) complexity with respect to the number of frames, enabling efficient long video generation with minimal quality loss. The method is validated on several state-of-the-art video generation models (HunyuanVideo, Wan2.1-14B, Mochi 1), demonstrating significant speedups in both training and inference, especially for long videos, and supports efficient LoRA-based fine-tuning.

Strengths
- Principled Motivation: The identification and quantitative validation of the spatiotemporal energy decay phenomenon provides a strong, intuitive basis for the proposed sparsity pattern.
- Technical Innovation: The static, exponentially-decaying attention mask is both hardware-friendly and effective, achieving substantial computational savings without sacrificing video quality.
- Comprehensive Evaluation: Extensive experiments across multiple large-scale models and settings (default and extended video lengths) demonstrate consistent speedups (up to 3.7× inference, 4.4× training) and quality preservation.
- Ablation and Robustness: The authors provided additional ablation studies and quantitative analyses in the rebuttal, addressing concerns about design choices and the generality of the energy decay phenomenon across models and complex scenarios.

Weaknesses
- Initial Lack of Quantitative Analysis: The paper relied on qualitative evidence for the energy decay phenomenon. The rebuttal addressed this with detailed regression analyses across diverse prompts and models, showing strong quantitative support.
- Ablation Studies: Reviewers requested more ablations on mask design choices. The authors provided these in the rebuttal, validating their default settings.
- Quadratic Complexity in Spatial Resolution: While Aura Attention is O(n log n) in frames, it remains quadratic in spatial resolution. The authors clarified that practical speedups are maintained at higher resolutions and discussed how spatial compression techniques can be combined with their method.
- Baselines and Comparisons: Some reviewers suggested including more linear attention baselines. The authors added results for additional baselines in the rebuttal, showing that Aura Attention outperforms them in the video setting.

Overall, the paper received two clear accept ratings and two borderline accept ratings, with all reviewers acknowledging the technical soundness, empirical rigor, and potential impact of the work. ACs concur with the reviewer consensus.

Suggestions for Camera-Ready:
- Incorporate the additional quantitative analyses and ablation studies from the rebuttal into the main paper.
- Further clarify the details of how attention decay is measured and visualized.
- Address minor presentation issues and typos as noted by reviewers.